# Convergence Rates of Constrained Expected Improvement

**Haowei Wang**
National University of Singapore
Singapore
haowei_wang@u.nus.edu

**Jingyi Wang**
Lawrence Livermore National Laboratory
Livermore, CA 94550
wang125@llnl.gov

**Nai-Yuan Chiang**
Lawrence Livermore National Laboratory
Livermore, CA 94550
chiang7@llnl.gov

**Zhongxiang Dai**
The Chinese University of Hong Kong, Shenzhen
China
daizhongxiang@cuhk.edu.cn

**Szu Hui Ng**
National University of Singapore
Singapore
isensh@nus.edu.sg

**Cosmin G. Petra**
Lawrence Livermore National Laboratory
Livermore, CA 94550
petra1@llnl.gov

## Abstract

Constrained Bayesian optimization (CBO) methods have seen significant success in black-box optimization with constraints. One of the most commonly used CBO methods is the constrained expected improvement (CEI) algorithm. CEI is a natural extension of expected improvement (EI) when constraints are incorporated. However, the theoretical convergence rate of CEI has not been established. In this work, we study the convergence rate of CEI by analyzing its simple regret upper bound. First, we show that when the objective function $f$ and constraint function $c$ are assumed to each lie in a reproducing kernel Hilbert space (RKHS), CEI achieves the convergence rates of $\mathcal{O}\left(t^{-\frac{1}{2}}\log^{\frac{d+1}{2}}(t)\right)$ and $\mathcal{O}\left(t^{\frac{-\nu}{2\nu+d}}\log^{\frac{\nu}{2\nu+d}}(t)\right)$ for the commonly used squared exponential and Matérn kernels ($\nu > \frac{1}{2}$), respectively. Second, we show that when $f$ is assumed to be sampled from Gaussian processes (GPs), CEI achieves similar convergence rates with a high probability. Numerical experiments are performed to validate the theoretical analysis.

## 1 Introduction

Bayesian optimization (BO) is an efficient method for optimizing expensive black-box functions without derivatives. It leverages probabilistic surrogate models, most commonly Gaussian processes (GPs), to balance exploration and exploitation in the search for optimal solutions [Frazier, 2018]. BO has found widespread success in diverse fields such as structural design [Mathern et al., 2021], machine learning hyperparameter tuning [Wu et al., 2019], robotics [Calandra et al., 2016], fusion design [Wang et al., 2024], etc.

While traditional BO is typically applied to unconstrained settings, many real-world problems involve black-box constraints that must be satisfied. This has motivated growing interest in constrained

Bayesian optimization (CBO), where surrogate models are also constructed for constraint functions [Bernardo et al., 2011] that are complex and expensive to evaluate, making CBO especially valuable in applications like engineering design [Song et al., 2024] and automated machine learning [Ungredda and Branke, 2024]. One of the very key difference between unconstrained and constrained optimization is that the feasible region for constrained optimization problem consists of the search space where all constraints must be satisfied. A general form of the constrained BO problem is:

$$\underset{\boldsymbol{x} \in C}{\text{minimize}} \quad f(\boldsymbol{x}), \quad \text{subject to} \quad c(\boldsymbol{x}) \leq 0, \tag{1}$$

where $f : \mathbb{R}^d \to \mathbb{R}$ is the objective function, and $c : \mathbb{R}^d \to \mathbb{R}^m$ are the constraint functions. Both are defined on a compact input space $C \subset \mathbb{R}^d$. The objective and the constraint functions are both expensive black-box functions, that can only be evaluated through expensive physical or computer experiments. Throughout this paper, we consider the noise-free setting for both the objective and the constraints, *i.e.*, the function evaluations are deterministic and the true function values can be observed (see Remark 3.14 for discussion on the noisy case). In addition, a single constraint is considered, *i.e.*, $m = 1$, for simplicity of presentation. We note that our analysis can be easily extended to multiple constraints (see Remark 3.13 for details).

Broadly, CBO methods can be categorized into implicit and explicit approaches [Amini et al., 2025]. Implicit methods modify standard acquisition functions to incorporate constraints via merit functions or feasibility weights. Explicit methods estimate the feasible region directly and restrict search to this region. Among these, the constrained expected improvement (CEI) [Schonlau et al., 1998, Gelbart et al., 2014, Gardner et al., 2014] stands out as one of the most basic and widely adopted methods. CEI is a natural extension of the well-known expected improvement (EI) function [Jones et al., 1998], where the acquisition function is computed as the product of EI and the probability of feasibility. Thanks to this simple and interpretable formulation, CEI has been successfully applied across domains, and it remains one of the default choices in many constrained BO software packages [Balandat et al., 2020].

Despite its empirical popularity, the theoretical understanding of CEI lags behind. In contrast, unconstrained EI has been more extensively studied. Under a frequentist assumption where the objective $f$ lies in a reproducing kernel Hilbert space (RKHS), Bull [2011] established the convergence rate of EI by deriving the simple regret upper bound. Other works explored the density of sampled sequences [Vazquez and Bect, 2010] or connections between EI and optimal computing budget allocation [Ryzhov, 2016]. However, convergence rates (*i.e.*, simple regret upper bound) for CEI have not been rigorously established—neither under frequentist nor under Bayesian settings. Here, Bayesian setting means the objective $f$ is a function sampled from a GP.

Introducing constraints into EI significantly complicates the theoretical analysis. Unlike in the unconstrained case, the algorithm may need to explore infeasible regions to gain information on the constraint boundary. Furthermore, CEI's acquisition function is inherently more complex and non-convex, posing challenges for analysis. On the other hand, the presence of constraints in CEI leads to changes in the sampling procedure. As a result, the key challenge to study the convergence rate of CEI lies in analyzing the exploration (searching for feasible regions) and exploitation (optimizing within feasible areas) since the feasibility threshold is unknown in the input space.

In this paper, we provide the first theoretical convergence rates for CEI, focusing on simple regret upper bounds under both the frequentist and Bayesian settings. Our convergence rates provide practitioners theoretical assurance for the practical deployment of CEI. We explain the technical challenges and how we address them in Section 3. Our contributions are summarized as follows:

- Under the frequentist setting, we derive simple regret upper bounds of $\mathcal{O}\left(t^{-\frac{1}{2}} \log^{\frac{d+1}{2}}(t)\right)$ for the squared exponential (SE) kernel and $\mathcal{O}\left(t^{\frac{-\nu}{2\nu+d}} \log^{\frac{\nu}{2\nu+d}}(t)\right)$ for Matérn kernels ($\nu > \frac{1}{2}$). These bounds are improved upon the direct extension of Bull [2011] to the constrained case for SE kernel with $d \geq 3$ and Matérn kernels with $d \geq 3, \nu \geq \frac{d}{d-2}$. (see Theorem 3.7).
- Under the Bayesian setting for the objective, we achieve similar simple regret upper bounds with high probabilities. These bounds are established based on the newly derived bounds (see Theorem 3.11) on the difference between the improvement function and its corresponding EI in the Bayesian setting.

This paper is organized as follows. In Section 2, we describe the basics and preliminaries of BO, including the CEI algorithm. In Section 3, the simple regret upper bounds of CEI are established in both settings. Numerical experiments to validate the theoretical results are given in Section 4. Conclusions are made in Section 5. All proof details are presented in the appendix.

## 2 Background

CBO mainly consists of two components: the GP surrogates for the black-box objective function $f$ and constraint function $c$, and the constrained acquisition function as the sequential sampling rule guiding for the global optimum.

### 2.1 Gaussian process models for $f$ and $c$

Without losing generality, let the mean function for the objective GP model prior be $0$ and the covariance function (kernel) be $k_f(\boldsymbol{x}, \boldsymbol{x}') : \mathbb{R}^n \times \mathbb{R}^n \to \mathbb{R}$. At sample point $\boldsymbol{x}_t \in C$, we denote the objective function value as $f(\boldsymbol{x}_t)$ and the observed constraint function value is $c(\boldsymbol{x}_t)$. Given $t$ sample points, denote $\boldsymbol{x}_{1:t} = [\boldsymbol{x}_1, \ldots, \boldsymbol{x}_t]$ and $\boldsymbol{f}_{1:t} = [f(\boldsymbol{x}_1), \ldots, f(\boldsymbol{x}_t)]$. Moreover, denote the $t \times t$ covariance matrix $\boldsymbol{K}_t^f = [k_f(\boldsymbol{x}_1, \boldsymbol{x}_1), \ldots, k_f(\boldsymbol{x}_1, \boldsymbol{x}_t); \ldots; k_f(\boldsymbol{x}_t, \boldsymbol{x}_1), \ldots, k_f(\boldsymbol{x}_t, \boldsymbol{x}_t)]$. The posterior distribution of $f(\boldsymbol{x})|\boldsymbol{x}_{1:t}, \boldsymbol{f}_{1:t} \sim \mathcal{N}(\mu_t^f(\boldsymbol{x}), (\sigma_t^f(\boldsymbol{x}))^2)$ can then be inferred using Bayes' rule as follows

$$
\begin{aligned}
\mu_t^f(\boldsymbol{x}) &= (\boldsymbol{k}_t^f(\boldsymbol{x}))^T (\boldsymbol{K}_t^f)^{-1} \boldsymbol{f}_{1:t}, \\
(\sigma_t^f)^2(\boldsymbol{x}) &= k_f(\boldsymbol{x}, \boldsymbol{x}) - (\boldsymbol{k}_t^f(\boldsymbol{x}))^T (\boldsymbol{K}_t^f)^{-1} \boldsymbol{k}_t^f(\boldsymbol{x}),
\end{aligned}
\tag{2}
$$

where $\boldsymbol{k}_t^f(\boldsymbol{x}) = [k_f(\boldsymbol{x}_1, \boldsymbol{x}), \ldots, k_f(\boldsymbol{x}_t, \boldsymbol{x})]^T$. Similarly, denote the kernel for $c$ as $k_c : \mathbb{R}^n \times \mathbb{R}^n \to \mathbb{R}$ and the covariance matrix $\boldsymbol{K}_t^c = [k_c(\boldsymbol{x}_1, \boldsymbol{x}_1), \ldots, k_c(\boldsymbol{x}_1, \boldsymbol{x}_t); \ldots; k_c(\boldsymbol{x}_t, \boldsymbol{x}_1), \ldots, k_c(\boldsymbol{x}_t, \boldsymbol{x}_t)]$. The posterior distribution for $c$ is

$$
\begin{aligned}
\mu_t^c(\boldsymbol{x}) &= (\boldsymbol{k}_t^c(\boldsymbol{x}))^T (\boldsymbol{K}_t^c)^{-1} \boldsymbol{c}_{1:t}, \\
(\sigma_t^c)^2(\boldsymbol{x}) &= k_c(\boldsymbol{x}, \boldsymbol{x}) - (\boldsymbol{k}_t^c(\boldsymbol{x}))^T (\boldsymbol{K}_t^c)^{-1} \boldsymbol{k}_t^c(\boldsymbol{x}),
\end{aligned}
$$

where $\boldsymbol{k}_t^c(\boldsymbol{x}) = [k_c(\boldsymbol{x}_1, \boldsymbol{x}), \ldots, k_c(\boldsymbol{x}_t, \boldsymbol{x})]^T$, and $\mu_t^c(\boldsymbol{x})$ and $(\sigma_t^c)^2(\boldsymbol{x})$ are the posterior mean and variance for $c$, respectively. Here we use the subscripts $_f$, $_c$ and superscripts $^f$, $^c$ to distinguish between GPs for $f$ and $c$. Choices of the kernels $k_f$ and $k_c$ include the SE and Matérn kernels, which are among the most popular kernels for GP and BO. Their definitions are as follows.

$$
k_{SE}(\boldsymbol{x}, \boldsymbol{x}') = \exp\left(-\frac{r^2}{2l^2}\right), \ k_{Matérn}(\boldsymbol{x}, \boldsymbol{x}') = \frac{1}{\Gamma(\nu)2^{\nu-1}} \left(\frac{\sqrt{2\nu}r}{l}\right)^\nu B_\nu \left(\frac{\sqrt{2\nu}r}{l}\right),
$$

where $l > 0$ is the length hyper-parameters, $r = \|\boldsymbol{x} - \boldsymbol{x}'\|_2$, $\nu > 0$ is the smoothness parameter of the Matérn kernel, and $B_\nu$ is the modified Bessel function of the second kind.

### 2.2 Constrained Expected Improvement

Acquisition functions are critical to the performances of BO algorithms. In the unconstrained setting, one of the most widely adopted acquisition functions is EI [Jones et al., 1998]. Given $t$ samples, the improvement function of $f$ used in EI is defined as

$$
I_t^f(\boldsymbol{x}) = \max\{f_t^+ - f(\boldsymbol{x}), 0\},
\tag{3}
$$

where $f_t^+ = \min_{i=1,\ldots,t} f(\boldsymbol{x}_i)$. The expectation of (3) conditioned on existing samples is EI, which has a closed form Brochu et al. [2010]:

$$
EI_t^f(\boldsymbol{x}) = (f_t^+ - \mu_t^f(\boldsymbol{x}))\Phi(z_t^f(\boldsymbol{x})) + \sigma_t^f(\boldsymbol{x})\phi(z_t^f(\boldsymbol{x})),
\tag{4}
$$

where $z_t^f(\boldsymbol{x}) = \frac{f_t^+ - \mu_t^f(\boldsymbol{x})}{\sigma_t^f(\boldsymbol{x})}$. The functions $\phi$ and $\Phi$ are the probability density function (PDF) and the cumulative distribution function (CDF) of the standard normal distribution, respectively. The $t + 1$th sample using EI is chosen by

$$
\boldsymbol{x}_{t+1} = \operatorname*{argmax}_{\boldsymbol{x} \in C} EI_t^f(\boldsymbol{x}).
\tag{5}
$$

Taking into account the constraint, the constrained improvement function in CEI [Gardner et al., 2014] is defined as

$$I_t^C = \Delta_t^c(\boldsymbol{x})\max\{f_t^+ - f(\boldsymbol{x}), 0\}, \tag{6}$$

where $\Delta_t^c \in \{0, 1\}$ is the feasibility indicator function where $\Delta_t^c(\boldsymbol{x}) = 1$ if $c(\boldsymbol{x}) \leq 0$ and $\Delta_t^c(\boldsymbol{x}) = 0$ otherwise. The incumbent $f_t^+$ in CEI is augmented to be the best feasible observation. CEI assumes that $f$ and $c$ are conditionally independent [Gardner et al., 2014]. Taking the conditional expectation of (6), the CEI function is

$$EI_t^C(\boldsymbol{x}) = P_t(\boldsymbol{x})EI_t^f(\boldsymbol{x}) = \Phi\left(-\frac{\mu_t^c(\boldsymbol{x})}{\sigma_t^c(\boldsymbol{x})}\right)EI_t^f(\boldsymbol{x}), \tag{7}$$

where $P_t(\cdot)$ is the probability of feasibility (POF) function for $c(\boldsymbol{x}) \leq 0$. CEI chooses the next sample via

$$\boldsymbol{x}_{t+1} = \underset{\boldsymbol{x} \in C}{\arg\max}\, P_t(\boldsymbol{x})EI_t^f(\boldsymbol{x}). \tag{8}$$

The CEI algorithm is given in Algorithm 1.

---

**Algorithm 1** CEI algorithm

---

1: Choose $k_f(\cdot, \cdot)$, $k_c(\cdot, \cdot)$, and $T_0$ initial samples $\boldsymbol{x}_i, i = 1, \ldots, T_0$. Observe $\boldsymbol{f}_{1:T_0}$ and $\boldsymbol{c}_{1:T_0}$.
2: Train the GP surrogate models for $f$ and $c$ respectively conditioned on the initial observations.
3: **for** $t = T_0 + 1, T_0 + 2, \ldots$ **do**
4:   Find $\boldsymbol{x}_{t+1}$ based on (8) (CEI).
5:   Observe $f(\boldsymbol{x}_{t+1})$ and $c(\boldsymbol{x}_{t+1})$.
6:   Update the GP models with the addition of $\boldsymbol{x}_{t+1}$, $f(\boldsymbol{x}_{t+1})$, and $c(\boldsymbol{x}_{t+1})$.
7:   **if** Evaluation budget exhausted **then**
8:     Exit

---

CEI can be extended to multiple constraints assuming conditional independence among the constraints [Gardner et al., 2014]. Our derived convergence rates can also be readily extended to multiple constraints, as we explain in Remark 3.13.

## 3 Convergence rates of CEI

We present our main results of convergence rates for CEI by establishing the simple regret upper bounds. Denote the optimal solution to the constrained optimization problem (1) as $\boldsymbol{x}^*$. In the unconstrained case, the simple regret of EI is defined as $f_t^+ - f(\boldsymbol{x}^*)$ [Bull, 2011]. In the constrained case, we use the current best feasible observation and compare it to the optimal solution $f(\boldsymbol{x}^*)$, since one could have an infeasible sample point with smaller objective than $f(\boldsymbol{x}^*)$. Given that $f_t^+$ is already defined as the best feasible observation till iteration $t$ in CEI, we continue to use

$$r_t = f_t^+ - f(\boldsymbol{x}^*), \tag{9}$$

as the simple regret for CEI. In our analysis, we make the same underlying assumption as CEI that $f_t^+$ exists. In the following, we first establish the convergence rate under the frequentist assumptions in Section 3.1, including an improved version of the rate under frequentist assumptions in Section 3.1.1. Then, we establish the convergence rate under Bayesian objective assumptions in Section 3.2.

### 3.1 Simple regret upper bound under frequentist assumptions

In this section, we present the simple regret upper bound for CEI under the frequentist setting. Moreover, by adopting the information theory-based bounds and techniques in the noise-free cumulative regret bound of upper confidence bound (UCB) [Lyu et al., 2019], we can derive an improved upper bound in some cases compared to Bull [2011]. The definition of RKHS is given below.

**Definition 3.1.** Let $k$ be a positive definite kernel $k : \mathcal{X} \times \mathcal{X} \to \mathbb{R}$ with respect to a finite Borel measure supported on $\mathcal{X}$. A Hilbert space $H_k$ of functions on $\mathcal{X}$ with an inner product $\langle \cdot, \cdot \rangle_{H_k}$ is

called a RKHS with kernel $k$ if $k(\cdot, \boldsymbol{x}) \in H_k$ for all $\boldsymbol{x} \in \mathcal{X}$, and $\langle f, k(\cdot, \boldsymbol{x}) \rangle_{H_k} = f(\boldsymbol{x})$ for all $\boldsymbol{x} \in \mathcal{X}, f \in H_k$. The induced RKHS norm $\|f\|_{H_k} = \sqrt{\langle f, f \rangle_{H_k}}$ measures the smoothness of $f$ with respect to $k$.

In this section, we assume the following assumptions on the functions $f$ and $c$.

**Assumption 3.2.** The functions $f$ and $c$ lie in the RKHS, denoted as $\mathcal{H}_k^f(C)$ and $\mathcal{H}_k^c(C)$ associated with their respective bounded kernel $k_f$ and $k_c$, with the norm $\|\cdot\|_{H_k^f}$ and $\|\cdot\|_{H_k^c}$. The kernels satisfy $k_f(\boldsymbol{x}, \boldsymbol{x}') \leq 1, k_c(\boldsymbol{x}, \boldsymbol{x}') \leq 1, k_f(\boldsymbol{x}, \boldsymbol{x}) = 1$, and $k_c(\boldsymbol{x}, \boldsymbol{x}) = 1$, for $\forall \boldsymbol{x}, \boldsymbol{x}' \in C$. The RKHS norms are bounded above by constants $B_f$ and $B_c$, respectively, *i.e.*, $\|f\|_{H_k^f} \leq B_f, \|c\|_{H_k^c} \leq B_c$. Moreover, the bound constraints set $C$ is compact.

**Technical Challenges under Frequentist Assumptions.** The main challenge in establishing a simple regret upper bound for CEI is how to incorporate the constraint $c(\boldsymbol{x}) \leq 0$ and the probability of feasibility function $P_t(\boldsymbol{x})$ into the analysis. Existing regret bounds analysis on CBO methods often focus on UCB-type methods [Lu and Paulson, 2022, Zhou and Ji, 2022], for which the acquisition functions do not have the multiplicative structure between the objective and the constraint.

Under Assumption 3.2, both $f$ and $c$ are bounded on $C$ by their RKHS norm bounds, as stated in Lemma B.1. The simple regret upper bound is given in the following theorem.

**Theorem 3.3.** *Under Assumption 3.2, the CEI algorithm leads to the simple regret upper bound of*

$$r_t \leq \frac{c_{\tau B}}{\Phi(-B_c)} \left[ B_f \frac{4}{t-2} + (0.4 + B_f)\sigma_{t_k}^f(\boldsymbol{x}_{t_k+1}) \right], \tag{10}$$

*for some $t_k \in [\frac{t}{2} - 1, t]$, and $c_{\tau B} = \frac{\tau(B_f)}{\tau(-B_f)}$.*

**Sketch of Proof for Theorem 3.3.** We start by noticing that the sum of the difference between consecutive best feasible observations is bounded, *i.e.*, $\sum_{t=1}^{T} f_{t-1}^+ - f_t^+ \leq 2B_f$. Then, we adopt a technique in Bull [2011] to find $t_k$ such that $f_{t_k}^+ - f_{t_k+1}^+ \leq \frac{2B_f}{k}$, where $k \leq t_k \leq 2k$ and $2k \leq t \leq 2(k+1)$. Next, using the monotonicity of $f_t^+$, $r_t$ is bounded by $r_{t_k}$. Using the inequality between $I_t^f$ and $EI_t^f$ in Lemma B.4, we can bound $r_{t_k}$ by the EI on objective: $EI_{t_k}^f(\boldsymbol{x}^*)$. Then, we transform $EI_{t_k}^f(\boldsymbol{x}^*)$ into $EI_{t_k}^f(\boldsymbol{x}_{t_k+1})$ by inserting the term $P_{t_k}(\boldsymbol{x}^*)$, taking advantage of the multiplicative structure of CEI. The upper bound of $r_t$ then consists of the term $\frac{1}{P_{t_k}(\boldsymbol{x}^*)} EI_{t_k}^f(\boldsymbol{x}_{t_k+1})$. From the confidence interval $|f(\boldsymbol{x}) - \mu_t^f(\boldsymbol{x})|$ (Lemma B.2) and the fact that $f_{t_k}^+ - f_{t_k+1}^+ \leq \frac{2B_f}{k}$, we can bound $EI_{t_k}^f(\boldsymbol{x}_{t_k+1})$. The constraint term $\frac{1}{P_{t_k}(\boldsymbol{x}^*)}$ remains to be bounded. We use the confidence interval on $|c(\boldsymbol{x}) - \mu_t^c(\boldsymbol{x})|$ in Lemma B.2 at $\boldsymbol{x}^*$ and the fact that $\boldsymbol{x}^*$ is a feasible solution to obtain a lower bound for $P_{t_k}(\boldsymbol{x}^*)$. This concludes the proof.

*Remark* 3.4 (Constraint in the simple regret upper bound). The terms derived from the constraint function in (10) is $\frac{1}{\Phi(-B_c)}$, which emerges from the probability of feasibility function and $\mu_t^c(\boldsymbol{x})$ and $\sigma_t^c(\boldsymbol{x})$ of the GP model of $c(\boldsymbol{x})$. Thanks to the multiplicative structure between the objective and constraint in $I_t^C$ (6) and $EI_t^c$ (7), the simple regret upper bound maintains a similar form.

It is clear from (10) that the convergence of $r_t$ relies on the posterior standard deviation $\sigma_t^f(\boldsymbol{x}_{t+1})$. Since $t_k$ increases with $t$, as $\sigma_t^f(\boldsymbol{x}_{t+1}) \to 0$, so does $\sigma_{t_k}^f(\boldsymbol{x}_{t_k+1})$. In the noise-free setting, the posterior variance can be bounded via the maximum distance between sample points and a given point. To obtain the rate of simple regret bound, we use Assumptions (1)-(4) in Bull [2011] and focus on squared exponential (SE) and Matérn kernels. Recall that the smoothness parameter of the Matérn kernel is $\nu > 0$. Both the SE and Matérn kernels satisfy Assumptions (1)-(4) in Bull [2011], with SE kernel obtained as $\nu \to \infty$. Further, define

$$\eta = \begin{cases} \alpha, & \nu \leq 1 \\ 0, & \nu > 1, \end{cases} \tag{11}$$

where $\alpha = \frac{1}{2}$ if $\nu \in \mathbb{N}$, and $\alpha = 0$ otherwise. Then, for SE and Matérn kernels, $\sigma_{t_k}^f(\boldsymbol{x}_{t_k+1})$ can be bounded with the following lemma.

**Lemma 3.5** (Bull [2011]). *For the SE kernel, there exists constant $C' > 0$ so that given $\forall t \in \mathbb{N}$,*

$$\sigma_i^f(\boldsymbol{x}_{i+1}) \geq C' k^{-\frac{1}{d}} \tag{12}$$

*holds for at most $k$ times, for $\forall k \in \mathbb{N}$, $k \leq t$ and $i = 1, \ldots, t-1$. For Matérn kernels,*

$$\sigma_i^f(\boldsymbol{x}_{i+1}) \geq C' k^{-\frac{\min\{\nu,1\}}{d}} \log^\eta(k) \tag{13}$$

*holds at most $k$ times.*

In the constrained setting, we are able to obtain the same rates as those in the unconstrained case [Bull, 2011] using Lemma 3.5.

**Corollary 3.6.** *Under Assumption 3.2, the CEI algorithm leads to the convergence rates of*

$$\mathcal{O}\left(t^{-\frac{1}{d}}\right) \text{ and } \mathcal{O}\left(t^{-\frac{\min\{\nu,1\}}{d}} \log^\eta(t)\right), \tag{14}$$

*for SE and Matérn kernels, respectively, where $\eta$ is from* (11).

Corollary 3.6 shows that the CEI algorithm is guaranteed to find the best feasible point asymptotically with the rates elaborated in (14). Also, we point out that the choice of kernels and their parameters affect the convergence rates. Since the SE kernel can be viewed as a Matérn kernel with $\nu \to \infty$, its convergence rate is better than Matérn kernels with $\nu \leq 1$. However, due to the limitations of the kernel analysis in Bull [2011] (see Remark 3.8), for $\nu \geq 1$, SE and Matérn kernels have similar convergence rates in Corollary 3.6. As we present in the following section, improved rates for both kernels can be obtained in some cases.

### 3.1.1 Improved simple regret upper bound under frequentist assumptions

Next, we apply maximum information gain and the corresponding information theory to obtain improved simple regret upper bounds.

**Theorem 3.7.** *Under Assumption 3.2, the CEI algorithm leads to the improved convergence rates of*

$$\mathcal{O}\left(t^{-\frac{1}{2}} \log^{\frac{d+1}{2}}(t)\right) \text{ and } \mathcal{O}\left(t^{\frac{-\nu}{2\nu+d}} \log^{\frac{\nu}{2\nu+d}}(t)\right), \tag{15}$$

*for SE and Matérn kernels, respectively.*

**Sketch of Proof for Theorem 3.7.** The proof follows similar steps to that of Theorem 3.3 but further bounds $\sigma_{t_k}^f(\boldsymbol{x}_{t_k+1})$ using $\gamma_t^f$. To do so, we first recognize that the bound using $\gamma_t^f$ (Lemma A.3) is established in the noisy case where the posterior standard deviation has a different form as in (18). Using Lemma A.4, we can establish that the noise-free posterior standard deviation also satisfies $\sum_{i=0}^{t-1} \sigma_i^f(\boldsymbol{x}_{i+1}) \leq \sqrt{C_\gamma t \gamma_t^f}$. Then, from Lemma A.5, we can find a small enough $\sigma_i^f(\boldsymbol{x}_{i+1})$. Specifically, choose $k = [t/3]$, where $[x]$ denotes the largest integer smaller than $x$. Thus, we have $3k \leq t \leq 3(k+1)$. Then, there exists $k \leq t_k \leq 3k$ such that $f_{t_k}^+ - f_{t_k+1}^+ \leq \frac{2B_f}{k}$ and $\sigma_{t_k}^f(\boldsymbol{x}_{t_k+1}) \leq \frac{\sqrt{t\gamma_t^f}}{k}$. The rest of the proof follows from that of Theorem 3.7.

*Remark* 3.8 (Improved rate of convergence). As mentioned above, the rates in Corollary 3.6 are the same as the known convergence rates for EI in Bull [2011]. Meanwhile, the rates in Theorem 3.7 is an improvement over those of Bull [2011] for SE kernel with $d \geq 3$ and Matérn kernels with $d \geq 3, \nu \geq \frac{d}{d-2}$. To achieve this, we applied techniques from regret bound analysis on noise-free UCB [Lyu et al., 2019] that allows us to use maximum information gain to bound the sum of $\sigma_t^f(\boldsymbol{x}_{t+1})$. Then, we use our techniques in the proof of Theorem 3.3 to bound an individual $\sigma_{t_k}^f(\boldsymbol{x}_{t_k+1})$. In Bull [2011], the $\sigma_{t_k}^f(\boldsymbol{x}_{t_k+1})$ is bounded by the Taylor expansion of the kernel functions. Therefore, the rates of decrease are limited to quadratic terms for both SE and Matérn kernels, since their Taylor expansions around 0 for $\|\boldsymbol{x} - \boldsymbol{x}'\|_2$ are quadratic at best. On the other hand, maximum information gain can lead to tight bounds on $\gamma_t^f$ that take advantages of the spectral properties of the kernels [Vakili et al., 2021, Iwazaki, 2025]. Hence, using $\gamma_t^f$ to bound $\sigma_{t_k}^f(\boldsymbol{x}_{t_k+1})$ can produce a faster rate. As the open question raised in Vakili [2022] gets answered, further improvement of the convergence rates is possible, *e.g.*, using techniques from Iwazaki [2025].

## 3.2 Simple regret upper bound under Bayesian objective assumption

In this section, we present the simple regret upper bound for CEI under the Bayesian objective assumptions. We again use the maximum information gain to derive the simple regret upper bound.

**Assumption 3.9.** The bound constraint set $C \subset [0, r]^d$ is compact and convex. The objective function $f$ is sampled from $GP(0, k_f(\boldsymbol{x}, \boldsymbol{x}'))$. Further, the objective function $f$ is assumed to be Lipschitz continuous (of 1-norm) with Lipschitz constant $L_f$ with probability $\geq 1 - da_f e^{L_f^2/b_f^2}$ for some constants $a_f > 0$ and $b_f > 0$. The kernels satisfy $k_f(\boldsymbol{x}, \boldsymbol{x}') \leq 1$ and $k_f(\boldsymbol{x}, \boldsymbol{x}) = 1$. The constraint function $c$ remains in the RKHS of $k_c$, similarly to the frequentist setting.

In the remaining of this section we will work under Assumption 3.9.

**Technical Challenges under Bayesian Assumptions.** In addition to the challenges in the frequentist setting, the bounds on EI in the Bayesian setting are not available in current literature, to the best of our knowledge. Starting from the confidence interval on $|f(\boldsymbol{x}) - \mu_t^f(\boldsymbol{x})|$, we derive the bounds on $|I_t^f(\boldsymbol{x}) - EI_t^f(\boldsymbol{x})|$ with high probability, an important step towards the bound on $r_t$. Noticeably, under the Bayesian setting, the bounds are satisfied with a given probability, *e.g.*, $1 - \delta$, where $\delta \in (0, 1)$.

The simple regret upper bound is given in the following theorem.

**Theorem 3.10.** *Let* $\beta = 2\log(6c_\alpha/\delta)$ *and* $\beta_t = 2\log(3\pi_t/\delta)$, *where* $c_\alpha = \frac{1+2\pi}{2\pi}$ *and* $\pi_t = \frac{\pi^2 t^2}{6}$. *Under Assumption 3.9, the CEI algorithm leads to the simple regret upper bound*

$$r_t \leq c_\tau(\beta) \frac{1}{\Phi(-B_c)} \left[ \frac{4M_f}{t-2} + \frac{2\beta_t^{1/2}}{t-2} \sqrt{C_\gamma t \gamma_t^f} + (0.4 + \beta^{1/2}) \sigma_{t_k}^f(\boldsymbol{x}_{t_k+1}) \right], \tag{16}$$

*for some* $t_k \in [\frac{t}{2} - 1, t]$, $c_\tau(\beta) = \frac{\tau(\beta^{1/2})}{\tau(-\beta^{1/2})}$, *and constant* $M_f > 0$ *with probability* $\geq 1 - \delta$.

The constant $M_f$ is from Lemma C.1. The convergence rate is given in the next theorem.

**Theorem 3.11.** *Let* $\beta = 2\log(6c_\alpha/\delta)$ *and* $\beta_t = 2\log(3\pi_t/\delta)$, *where* $c_\alpha = \frac{1+2\pi}{2\pi}$ *and* $\pi_t = \frac{\pi^2 t^2}{6}$. *Under Assumption 3.9, the CEI algorithm leads to the convergence rates of*

$$\mathcal{O}\left(t^{-\frac{1}{2}} \log^{\frac{d+2}{2}}(t)\right) \text{ and } \mathcal{O}\left(t^{\frac{-\nu}{2\nu+d}} \log^{\frac{2\nu+0.5d}{2\nu+d}}(t)\right), \tag{17}$$

*for SE and Matérn kernels, respectively, with probability* $\geq 1 - \delta$.

**Sketch of Proof for Theorem 3.10.** Recall that $f$ and $c$ are assumed conditionally independent in CEI. We start from the bound on the confidence interval for $f$: $|f(\boldsymbol{x}) - \mu_t^f(\boldsymbol{x})| \leq \beta^{1/2} \sigma_t^f(\boldsymbol{x})$, with probability $\geq 1 - \delta$, where $\beta = 2\log(1/\delta)$, as in Lemma C.2. The confidence interval of $c$ remains the same as in the frequentist setting. These are well-known results [Srinivas et al., 2009]. Then, we derive the subsequent bounds $|I_t^f(\boldsymbol{x}) - EI_t^f(\boldsymbol{x})| \leq \sqrt{\beta}\sigma_t^f(\boldsymbol{x})$, where $\beta = \max\{1.44, 2\log(c_\alpha/\delta)\}$ and $c_\alpha = \frac{1+2\pi}{2\pi}$ with probability $\geq 1 - \delta$ (Lemma C.5). Then, we prove the relationship in Lemma C.6 that $I_t^f(\boldsymbol{x}) \leq \frac{\tau(\sqrt{\beta})}{\tau(-\sqrt{\beta})} EI_t^f(\boldsymbol{x})$ with probability $\geq 1 - \delta$. We can now follow the general analysis framework in Section 3.1 and Theorem 3.3 to obtain the simple regret upper bound under Bayesian objective assumptions, while choosing $t_k$ with a more defined criterion.

*Remark* 3.12 (Comparison to the frequentist setting). Comparing Theorem 3.7 to Theorem 3.11, the convergence rates in the frequentist and Bayesian settings are the same except for a $\log^{1/2}(t)$ term. This is partially because simple regret focuses on the best feasible solution $f_t^+$ and thus many of the parameters in Theorem 3.3 and 3.10 do not depend on $t$.

*Remark* 3.13 (Multiple constraints). As mentioned in Section 1, our results can be readily applied to CEI with multiple constraints for both frequentist and Bayesian settings. Consider $m$ constraints $c_i(\boldsymbol{x}) \leq 0, i = 1, \ldots, m$. Assuming conditional independence of the constraints, the CEI function is $EI_t^C(\boldsymbol{x}) = \Pi_{i=1}^m P_t^i(\boldsymbol{x}) EI_t^f(\boldsymbol{x}) = \Pi_{i=1}^m \Phi\left(\frac{-\mu_t^{c_i}(\boldsymbol{x})}{\sigma_t^{c_i}(\boldsymbol{x})}\right) EI_t^f(\boldsymbol{x})$, where $P_t^i$ is the probability of feasibility function of constraint $c_i(\boldsymbol{x}) \leq 0$, and $\mu_t^{c_i}(\boldsymbol{x})$ and $\sigma_t^{c_i}(\boldsymbol{x})$ are the posterior mean and standard deviation for $c_i$, respectively. By making the assumption that each constraint function lies

in its corresponding RKHS of the kernel $k_{c_i}$, we have $|c_i(\boldsymbol{x}) - \mu_t^{c_i}(\boldsymbol{x})| \le B_{c_i}\sigma_t^{c_i}(\boldsymbol{x})$, where $B_{c_i}$ is the upper bound of RKHS norm associated with kernel $k_{c_i}$ and function $c_i$. We can then apply the analysis framework in this paper to obtain an upper bound similar to that of Theorem 3.3, where the term $\frac{1}{\Phi(-B_c)}$ is replaced with $\Pi_{i=1}^{m}\frac{1}{\Phi(-B_{c_i})}$. We note that in the Bayesian objective setting, to ensure probability $1 - \delta$, the parameter $\beta$ needs to increase with the number of constraints as well, *e.g.*, $\beta = 2\log((m+5)c_\alpha/\delta)$.

*Remark* 3.14 (Extension to the noisy setting). Extending our analysis to the noisy setting is non-trivial, and we discuss the associated challenges for noisy objective and constraint functions separately. A noisy constraint function introduces additional complications in defining feasibility. If only noisy observations of the constraint values are available, the notion of a feasible sample and the definition of $f_t^+$ becomes ambiguous. As a result, major modifications to the CEI algorithm are required to appropriately handle the uncertainty introduced by noise.

For the noisy objective function, CEI can be adapted similarly to the noisy EI formulation by treating the best feasible noisy observation as the incumbent. However, to the best of our knowledge, a theoretical guarantee on the simple regret bound for the noisy unconstrained setting remains unavailable. Recent work by Wang et al. [2025] provides a framework for deriving noisy simple regret bounds based on the best observed value, $r_t^s = y_t^+ - f(\boldsymbol{x}_t)$, which can be extended to CEI. Specifically, by defining $r_t^s$ as the simple regret for CEI with $y_t^+$ denoting the best feasible noisy observation, a similar proof strategy as in Theorem 3.10 yields an analogous upper bound. In the Bayesian setting with i.i.d. Gaussian noise on the objective and noise-free constraint observations, the convergence rate of the upper bound on $r_t^s$ can be obtained. However, we note that given the noise, $r_t^s$ is possibly negative.

*Remark* 3.15 (Infeasible initial sample). It is well known that CEI requires initial feasible sample [Gardner et al., 2014]. That is, $f_t^+$ exists from the initial samples so that the CEI calculation can proceed. Methods proposed to address this issue typically employ separate strategies when no feasible samples are available and revert to the standard CEI formulation once feasibility is established [Lin et al., 2024, Letham et al., 2019]. In addition, introducing a tolerance parameter in the constraint can further mitigate this problem by allowing near-feasible points when the degree of violation is small.

*Remark* 3.16 (Tolerance in constraints). In gradient-based optimization methods, a tolerance for constraint violation is often used to improve the performance and flexibility of algorithms Wächter and Biegler [2006], Nocedal and Wright [2006]. Motivated by this, we introduce a tolerance parameter $\lambda \ge 0$, where a point $\boldsymbol{x}$ is considered feasible if $c(\boldsymbol{x}) \le \lambda$ and infeasible otherwise. The corresponding CEI with tolerance is defined as $EI_t^C(\boldsymbol{x}, \lambda) = P_t(\boldsymbol{x}, \lambda)EI_t^f(\boldsymbol{x}) = \Phi\left(\frac{\lambda - \mu_t^c(\boldsymbol{x})}{\sigma_t^c(\boldsymbol{x})}\right)EI_t^f(\boldsymbol{x})$. Clearly, the standard CEI formulation is recovered when $\lambda = 0$.

The simple regret bound is affected by $\lambda$ and should lead to $\frac{1}{\Phi(\lambda - B_c)}$ in place of $\frac{1}{\Phi(-B_c)}$. In fact, we can replace $\frac{1}{\Phi(\lambda - B_c)}$ with $1/\Phi\left(\frac{\lambda}{\sigma_{t_k}^c(x^*)} - B_c\right)$, which is time-varying. One can follow the proof of Theorem 3.3 to obtain this term, which emerges from the confidence interval of $c$ at $t_k$, $x^*$ and $c(x^*) \le 0$.

As the sample iteration increases, the inclusion of $\sigma_{t_k}^c(x^*)$ is important in balancing $-B_c$ that can lead to a large simple regret upper bound. We explain the intuition below. As $t \to \infty$, $t_k \to \infty$ and $k \to \infty$. We know $\sigma_t^f(\boldsymbol{x}_{t+1}) \to 0$, and hence $\sigma_{t_k}^f(\boldsymbol{x}_{t_k+1}) \to 0$ and $r_t \to 0$. That is, the simple regret upper bound of CEI with $\lambda > 0$ converges. Thus, $\boldsymbol{x}_t$ approaches at least one of the optimal solutions. Suppose without losing generality, $\boldsymbol{x}_t \to \boldsymbol{x}^*$. Then, by definition $\sigma_{t_k}^c(\boldsymbol{x}^*) \to 0$. Consequently, we should have $\frac{\lambda}{\sigma_{t_k}^c(x^*)} \to \infty$ for $\lambda > 0$. Then, we have $\Phi\left(\frac{\lambda}{\sigma_{t_k}^c(x^*)}\right) \to 1$. Therefore, $1/\Phi(\frac{\lambda}{\sigma_{t_k}^c(x^*)} - B_c) \to 1$ and $B_c$ does not affect the simple regret upper bound asymptotically. We note that the convergence rate of CEI with tolerance remains similar since it is dominated by the maximum information gain of $f$.

# 4   Numerical experiments

Although this paper is primarily theoretical, we conduct numerical experiments to support the theoretical results. We apply the CEI algorithm to eight synthetic problems that are randomly generated from RKHS of kernels and GP priors, and five benchmark problems commonly used in the

CBO literature. These numerical experiments are not intended to demonstrate superior performance over the state-of-the-art CBO algorithms. Instead, they serve as empirical evidence for the theoretical analysis presented in this work. All experiments are conducted on M1 (16GB memory)[1].

## 4.1 Synthetic problems

In this section, we study objective and constraint functions drawn from reproducing kernel Hilbert spaces (RKHSs) as well as from Gaussian process priors with Matérn ($\nu = 2.5$) and squared exponential (SE) kernels, across input dimensions $d \in 2, 4$. The domain is the hypercube $[0, 1]^d$. For RKHS cases (the frequentist setting), the functions are generated with a similar approach to Chowdhury and Gopalan [2017]. Specifically, both objective $f(x)$ and constraint functions $c(x)$ are generated by sampling from the RKHS associated with a chosen kernel (Matérn/SE kernels with a length scale of $0.2$). Each function is constructed as a weighted sum of kernel evaluations at $100$ randomly selected basis points, with weights drawn from a standard normal distribution. Formally, the function takes the form $f(x) = \sum_{i=1}^{n} \alpha_i k(x, X_i)$ , where $k$ is the kernel, $X_i$ are basis points, and $\alpha_i$ are random coefficients; $c(x)$ is generated similarly. For the GP cases (the Bayesian setting), the functions are generated with an approach similar to Srinivas et al. [2009]. Specifically, we uniformly choose 1000 points in the design space and sample randomly from a multivariate Gaussian distribution defined by the GP prior with the chosen kernel.

For each synthetic problem, we conducted 100 independent trials. The number of initial design is set to $10d$, and 50 optimization iterations were performed for all cases. We plotted the log-log curve of simple regret against the number of iterations in Figure 1. In all cases, we consistently observed sublinear convergence patterns, which align well with our theoretical guarantees.

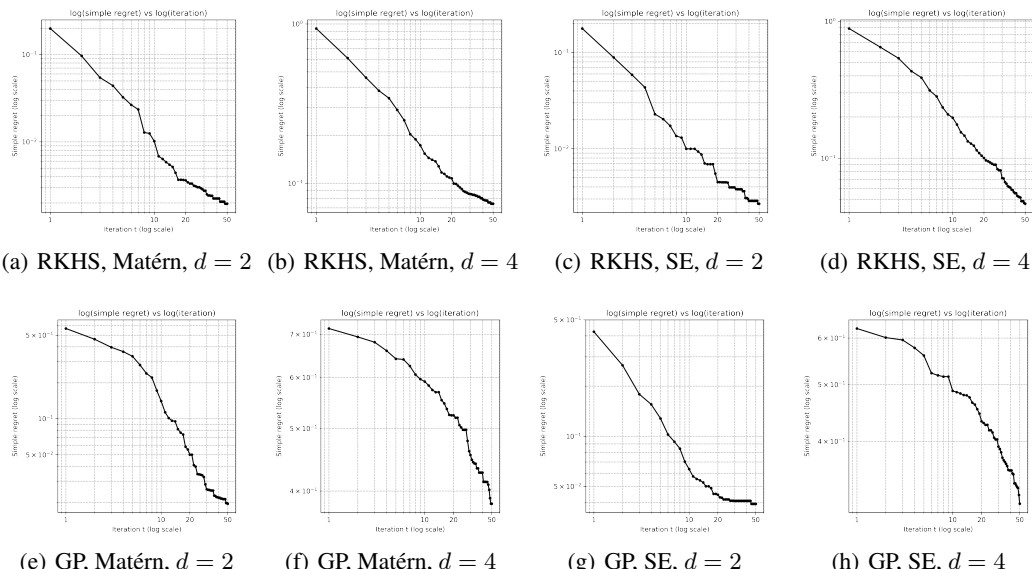

| (a) RKHS, Matérn, $d = 2$ | (b) RKHS, Matérn, $d = 4$ | (c) RKHS, SE, $d = 2$ | (d) RKHS, SE, $d = 4$ |

| (e) GP, Matérn, $d = 2$ | (f) GP, Matérn, $d = 4$ | (g) GP, SE, $d = 2$ | (h) GP, SE, $d = 4$ |

Figure 1: The log-log plots for simple regret vs optimization iterations of CEI for the synthetic problems.

## 4.2 Test problems

Next, we evaluate simple regret of CEI on five commonly used test problems in the literature of CBO. Specifically, Problem 1 tests performance in a small feasible region, which was previously studied in Gardner et al. [2014], Ariafar et al. [2019]. Problem 2 includes multiple constraints and local minimums, which has been used in Gramacy et al. [2016], Hernández-Lobato et al. [2015]. Problem 3 is a four-dimensional problem, previously studied in Picheny et al. [2016], Ariafar et al. [2019].

---

[1]Codes are available in https://github.com/Haowei-Wang/Convergence-Rates-of-Constrained-Expected-Improvement.

Problem 4 is the six-dimensional Hartmann problem, previously tested in Letham et al. [2019]. Problem 5 is the Rosenbrock function, where the global minimum lies in a narrow region. The mathematical formulations of the five functions are presented in Appendix D. For two-dimensional problems, we also include the contour plots of the objective and constraint functions in Appendix D. The SE kernel is used for the GP modeling (similar performance is observed for the Matérn kernel) and the hyper-parameters are estimated by a standard maximum likelihood method.

For each test problem, we conducted 100 independent trials with different random initial designs. When CEI fails to identify a feasible sample, we adopt the same heuristic strategy as in Letham et al. [2019]. The numerical results are summarized in Figure 2. The solid line represents the median of the simple regret, and the dotted lines represent the 25th percentile and 75th percentile of the simple regret, respectively. From the figures, we observe that CEI consistently reduces simple regret, aligning with the asymptotic convergence theories established in this paper. Accross all problems, the simple regret converges to 0 quickly. The 25th percentile and 75th percentile results demonstrate the good statistical properties of CEI.

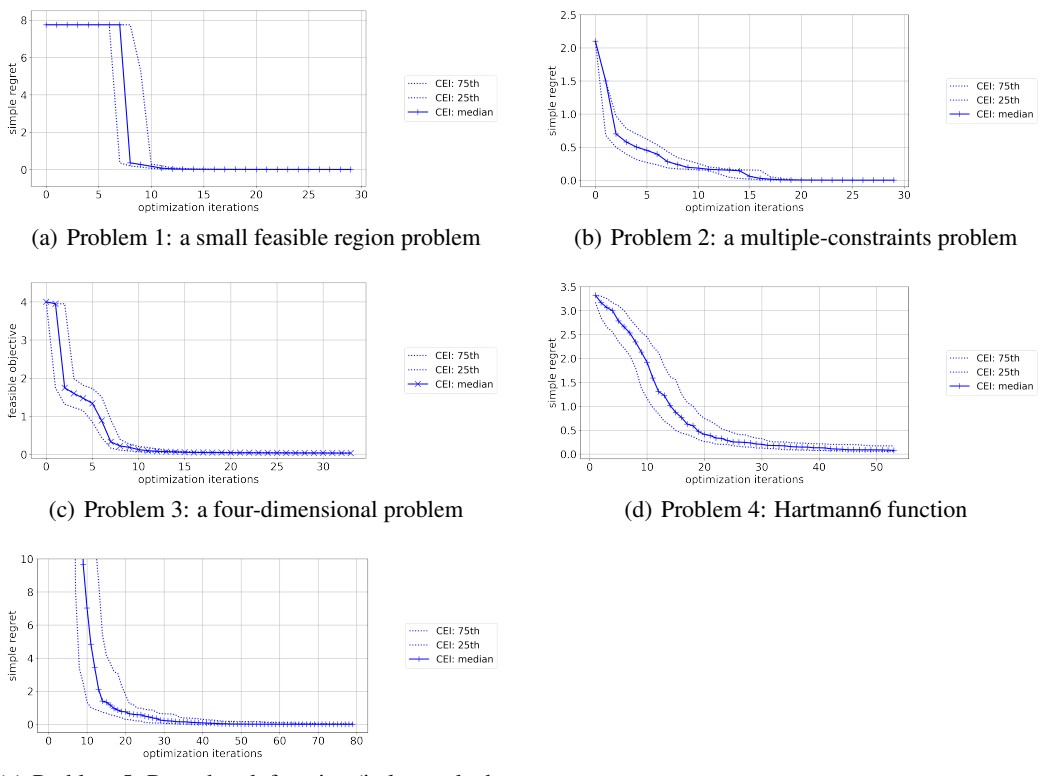

(a) Problem 1: a small feasible region problem     (b) Problem 2: a multiple-constraints problem

(c) Problem 3: a four-dimensional problem     (d) Problem 4: Hartmann6 function

(e) Problem 5: Rosenbrock function (in log scale due to large range of objective values.)

Figure 2: Simple regret of CEI for five test problems.

## 5 Conclusions

In this paper, we studied the simple regret upper bounds of the CEI algorithm, one of the most widely adopted CBO methods. Under both frequentist setting and Bayesian objective assumptions, we establish for the first time the convergence rates for CEI. Our results provide theoretical support and validation for the empirical success of CEI.

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

# A  Background information and preliminary results

## A.1  Information gain

To obtain state-of-the-art simple regret upper bound, we utilize the information theory results that are well-established in previous literature [27, 8, 30]. Using $f$ as the example, let $A \subset C$ denote a set of sampling points. Assume that the observations are noisy at sample points with $y_A = f(\boldsymbol{x}_A) + \epsilon_A$ at $\boldsymbol{x} \in A$, where $\epsilon_A \sim \mathcal{N}(0, \sigma^2)$ denotes the independent and identically distributed Gaussian noises. The maximum information gain is defined as follows.

**Definition A.1.** Given $\boldsymbol{x}_A$ and $\boldsymbol{y}_A$, the mutual information between $f$ and $\boldsymbol{y}_A$ is $I(\boldsymbol{y}_A; f_A) = H(\boldsymbol{y}_A) - H(\boldsymbol{y}_A | f_A)$, where $H$ denotes the entropy. The maximum information gain $\gamma_T^f$ after $T$ samples is $\gamma_T^f = \max_{A \subset C, |A| = T} I(\boldsymbol{y}_A; \boldsymbol{f}_A)$.

The rate of increase for $\gamma_t^f$ depends on the property of the kernel. For common kernels such as the SE kernel and the Matérn kernel, $\gamma_t^f$ has been studied in literature and the state-of-the-art rates of $\gamma_t^f$ are summarized in Lemma A.2 [30, 15].

**Lemma A.2.** *For GP with $t$ samples, the SE kernel has $\gamma_t = \mathcal{O}(\log^{d+1}(t))$, and the Matérn kernel with smoothness parameter $\nu > 0.5$ has $\gamma_t = \mathcal{O}(t^{\frac{d}{2\nu+d}}(\log^{\frac{2\nu}{2\nu+d}}(t)))$.*

The maximum information gain $\gamma_t^c$ for the constraint function can be defined similarly.

While $\gamma_t^f$ is defined in the noisy case, we can readily apply it to the noise-free case and bound $\sigma_t^f(\boldsymbol{x}_{t+1})$ using techniques similar to that in [20]. To do so, we note that given the Gaussian observation noise $\epsilon_t \sim \mathcal{N}(0, \sigma^2)$ in the GP model, the posterior prediction for $f$ becomes

$$
\begin{aligned}
\tilde{\mu}_t^f(\boldsymbol{x}) &= \boldsymbol{k}_t^f(\boldsymbol{x})(\boldsymbol{K}_t^f + \sigma^2 \boldsymbol{I})^{-1} \boldsymbol{f}_{1:t}, \\
(\tilde{\sigma}_t^f)^2(\boldsymbol{x}) &= k^f(\boldsymbol{x}, \boldsymbol{x}) - (\boldsymbol{k}_t^f)^T(\boldsymbol{x})(\boldsymbol{K}_t^f + \sigma^2 \boldsymbol{I})^{-1} \boldsymbol{k}_t^f(\boldsymbol{x}),
\end{aligned}
\tag{18}
$$

Similarly, we can define the posterior predictions for $c$ with noise in the GP model as $\tilde{\mu}_t^f(\boldsymbol{x})$ and $\tilde{\sigma}_t^f(\boldsymbol{x})$. The sum of posterior variance for GP generated by (2) satisfy the next lemma, based on information theory [27].

**Lemma A.3.** *The sum of GP posterior variances given $t$ samples satisfy*

$$
\sum_{t=1}^{T} \tilde{\sigma}_{t-1}^c(\boldsymbol{x}_t) \le \sqrt{C_\gamma T \gamma_T^c}, \ \sum_{t=1}^{T} \tilde{\sigma}_{t-1}^f(\boldsymbol{x}_t) \le \sqrt{C_\gamma T \gamma_T^f},
\tag{19}
$$

*where $C_\gamma = \frac{2}{\log(1+\sigma^{-2})}$ and $\gamma_t^f$ and $\gamma_t^c$ are the maximum information gains for $f$ and $c$, respectively.*

We have the following lemma for $\tilde{\sigma}_t^f(\boldsymbol{x})$ and $\sigma_t^f(\boldsymbol{x})$.

**Lemma A.4.** *The noise-free (GP) posterior standard deviation satisfies $\sigma_t^f(\boldsymbol{x}) < \tilde{\sigma}_t^f(\boldsymbol{x})$ for $\forall \sigma > 0$.*

*Proof.* We first note that all the eigenvalues of $\boldsymbol{K}_t^f$ is smaller than those of $\boldsymbol{K}_t^f + \sigma^2 \boldsymbol{I}$, since $\boldsymbol{K}_t^f$ is symmetric and positive definite. Thus,

$$
(\boldsymbol{k}_t^f)^T(\boldsymbol{x})(\boldsymbol{K}_t^f + \sigma^2 \boldsymbol{I})^{-1} \boldsymbol{k}_t^f(\boldsymbol{x}) < (\boldsymbol{k}_t^f)^T(\boldsymbol{x})(\boldsymbol{K}_t^f)^{-1} \boldsymbol{k}_t^f(\boldsymbol{x}),
\tag{20}
$$

for $\forall \sigma > 0$ and $\boldsymbol{k}_t^f(\boldsymbol{x})$. Therefore, by their definitions (2) and (18), the proof is complete. $\square$

The posterior standard deviation under assumptions (1)-(4) in [6] in the frequentist and noise-free setting is given in Lemma 3.5, whose proof is given below.

*Proof.* By Lemma 7 in [6], there exists $C' > 0$ so that

$$
\sigma_i^f(\boldsymbol{x}_{i+1}) \ge C' k^{-\frac{\min\{\nu, 1\}}{d}} \log^\eta(k),
\tag{21}
$$

at most $k$ times, for $\forall k \in \mathbb{N}$, $k \le t$, and $i = 1, \dots, t-1$. Therefore, for the SE kernel,

$$
\sigma_i^f(\boldsymbol{x}_{i+1}) \ge C' k^{-\frac{1}{d}},
\tag{22}
$$

at most $k$ times. For Matérn kernels, we have (21) at most $k$ times. $\square$

Using maximum information gain, a tighter bound on the posterior standard deviation can be obtained in the next lemma.

**Lemma A.5.** *Given $\forall t \in \mathbb{N}$ and $i = 1, 2, \ldots, t - 1$, for SE kernel, there exists constant $C' > 0$ so that*

$$\sigma_i^f(\boldsymbol{x}_{i+1}) \geq C' \frac{t^{\frac{1}{2}} \log^{\frac{d+1}{2}}(t)}{k}, \tag{23}$$

*holds for at most $k$ times, for $\forall k \leq t$. Similarly, for Matérn kernels with $\nu > 0.5$,*

$$\sigma_i^f(\boldsymbol{x}_{i+1}) \geq C' \frac{t^{\frac{\nu+d}{2\nu+d}} \log^{\frac{\nu}{2\nu+d}}(t)}{k}, \tag{24}$$

*holds at most $k$ times.*

*Proof.* From Lemma A.4 and Lemma A.3, we can write that

$$\sum_{i=1}^{t} \sigma_{i-1}^f(\boldsymbol{x}_i) \leq \sqrt{t\gamma_t^f}, \tag{25}$$

where we use without losing generality $C_\gamma \leq 1$. Therefore, for any $k \in \mathbb{N}$ and $k \leq t$,

$$\sigma_i^f(\boldsymbol{x}_{i+1}) \geq \frac{\sqrt{t\gamma_t^f}}{k}, \tag{26}$$

at most $k$ times. Therefore, for SE kernel, by Lemma A.2, there exists $C' > 0$ such that

$$\sigma_i^f(\boldsymbol{x}_{i+1}) \geq C' \frac{t^{\frac{1}{2}} \log^{\frac{d+1}{2}}(t)}{k}, \tag{27}$$

at most $k$ times. For Matérn kernels with $\nu > 0.5$,

$$\sigma_i^f(\boldsymbol{x}_{i+1}) \geq C' \frac{t^{\frac{\nu+d}{2\nu+d}} \log^{\frac{\nu}{2\nu+d}}(t)}{k}, \tag{28}$$

at most $k$ times. $\qquad\square$

Next, we state some basic properties of $\phi$, $\Phi$ and $\tau$ as a lemma.

**Lemma A.6.** *The PDF and CDF of standard normal distribution satisfy $0 < \phi(x) \leq \phi(0) < 0.4$, $\Phi(x) \in (0, 1)$, for any $x \in \mathbb{R}$. Given a random variable $\xi$ sampled from the standard normal distribution: $\xi \sim \mathcal{N}(0, 1)$, we have $\mathbb{P}\{\xi > c | c > 0\} \leq \frac{1}{2} e^{-c^2/2}$. Similarly, for $c < 0$, $\mathbb{P}\{\xi < c | c < 0\} \leq \frac{1}{2} e^{-c^2/2}$. The function $\tau(\cdot)$ is monotonically increasing.*

The last statement in Lemma A.6 is a well-known result (*e.g.*, see proof of Lemma 5.1 in [27]).

The next lemma proves basic properties for $EI_t^f$.

**Lemma A.7.** *For $\forall \boldsymbol{x} \in C$, $EI_t^f(\boldsymbol{x}) \geq 0$ and $EI_t^f(\boldsymbol{x}) \geq f_t^+ - \mu_t^f(\boldsymbol{x})$. Moreover,*

$$z_t^f(\boldsymbol{x}) \leq \frac{EI_t^f(\boldsymbol{x})}{\sigma_t^f(\boldsymbol{x})} < \begin{cases} \phi(z_t^f(\boldsymbol{x})), & z_t^f(\boldsymbol{x}) < 0, \\ z_t^f(\boldsymbol{x}) + \phi(z_t^f(\boldsymbol{x})), & z_t^f(\boldsymbol{x}) \geq 0. \end{cases} \tag{29}$$

*Proof.* From the definition of $I_t^f$ and $EI_t^f$, $EI_t^f(\boldsymbol{x}) \geq 0$ and $EI_t^f(\boldsymbol{x}) \geq y_t^+ - \mu_t^f(\boldsymbol{x})$ follow immediately. By (4),

$$\frac{EI_t^f(\boldsymbol{x})}{\sigma_t^f(\boldsymbol{x})} = z_t^f(\boldsymbol{x})\Phi(z_t^f(\boldsymbol{x})) + \phi(z_t^f(\boldsymbol{x})). \tag{30}$$

If $z_t^f(\boldsymbol{x}) < 0$, or equivalently $f_t^+ - \mu_t^f(\boldsymbol{x}) < 0$, (30) leads to $\frac{EI_t^f(\boldsymbol{x})}{\sigma_t^f(\boldsymbol{x})} < \phi(z_t^f(\boldsymbol{x}))$. If $z_t^f(\boldsymbol{x}) \geq 0$, we can write $\frac{EI_t^f(\boldsymbol{x})}{\sigma_t^f(\boldsymbol{x})} < z_t^f(\boldsymbol{x}) + \phi(z_t^f(\boldsymbol{x}))$. The left inequality in (29) is an immediate result of $EI_t^f(\boldsymbol{x}) \geq f_t^+ - \mu_t^f(\boldsymbol{x})$. $\qquad\square$

# B  Proofs for simple regret upper bound under frequentist assumptions

We state the boundedness result as a Lemma for easy reference.

**Lemma B.1.** *Under Assumption 3.2, $|f(\boldsymbol{x})| \leq B_f$ and $|c(\boldsymbol{x})| \leq B_c$ for all $\boldsymbol{x} \in C$.*

*Proof.* By Assumption 3.2, we can write

$$|f(\boldsymbol{x})| \leq \|f\|_{H_k^f} k_f(\boldsymbol{x}, \boldsymbol{x}) \leq B_f. \tag{31}$$

Similarly,

$$|c(\boldsymbol{x})| \leq \|c\|_{H_k^c} k_c(\boldsymbol{x}, \boldsymbol{x}) \leq B_c. \tag{32}$$

□

The following Lemma is a well-established result [8].

**Lemma B.2.** *At any given $\boldsymbol{x} \in C$ and $t \in \mathbb{N}$, the confidence intervals satisfy*

$$|f(\boldsymbol{x}) - \mu_t^f(\boldsymbol{x})| \leq B_f \sigma_t^f(\boldsymbol{x}), |c(\boldsymbol{x}) - \mu_t^c(\boldsymbol{x})| \leq B_c \sigma_t^c(\boldsymbol{x}). \tag{33}$$

Next, we present a lemma on the relationship between $I_t^f$ and $EI_t^f$, previously seen in [6].

**Lemma B.3.** *At $\boldsymbol{x} \in C, t \in \mathbb{N}$,*

$$I_t^f(\boldsymbol{x}) - EI_t^f(\boldsymbol{x}) \leq B_f \sigma_t^f(\boldsymbol{x}). \tag{34}$$

**Lemma B.4.** *The improvement function $I_t^f(\boldsymbol{x})$ and $EI_t^f(\boldsymbol{x})$ satisfy*

$$I_t^f(\boldsymbol{x}) \leq \frac{\tau(B_f)}{\tau(-B_f)} EI_t^f(\boldsymbol{x}), \tag{35}$$

*for $\forall \boldsymbol{x} \in C$ and $t \geq 1$.*

*Proof.* If $f_t^+ - f(\boldsymbol{x}) \leq 0$, then $I_t^f(\boldsymbol{x}) = 0$. Since $EI_t^f(\boldsymbol{x}) \geq 0$, (35) is trivial. If $f_t^+ - f(\boldsymbol{x}) > 0$, by Lemma B.2,

$$f_t^+ - \mu_t^f(\boldsymbol{x}) = f_t^+ - f(\boldsymbol{x}) + f(\boldsymbol{x}) - \mu_t^f(\boldsymbol{x}) > f(\boldsymbol{x}) - \mu_t^f(\boldsymbol{x}) > -B_f \sigma_t^f(\boldsymbol{x}). \tag{36}$$

From the monotonicity of $\tau$, we have

$$\tau\left(\frac{f_t^+ - \mu_t^f(\boldsymbol{x})}{\sigma_t^f(\boldsymbol{x})}\right) > \tau(-B_f), \tag{37}$$

and therefore,

$$EI_t^f(\boldsymbol{x}) = \sigma_t^f(\boldsymbol{x})\tau\left(\frac{f_t^+ - \mu_t^f(\boldsymbol{x})}{\sigma_t^f(\boldsymbol{x})}\right) > \tau(-B_f)\sigma_t^f(\boldsymbol{x}). \tag{38}$$

From Lemma B.3,

$$I_t^f(\boldsymbol{x}) - EI_t^f(\boldsymbol{x}) \leq B_f \sigma_t^f(\boldsymbol{x}). \tag{39}$$

Applying (39) to (38) leads to

$$EI_t^f(\boldsymbol{x}) > \frac{\tau(-B_f)}{B_f + \tau(-B_f)} I_t^f(\boldsymbol{x}) = \frac{\tau(-B_f)}{\tau(B_f)} I_t^f(\boldsymbol{x}). \tag{40}$$

□

Proof of Theorem 3.3 is given next.

*Proof.* From Lemma B.1,

$$\sum_{i=0}^{t-1} f_i^+ - f_{i+1}^+ = f_0^+ - f_t^+ \leq 2B_f. \tag{41}$$

Since $f_i^+ - f_{i+1}^+ \geq 0$, $f_i^+ - f_{i+1}^+ \geq \frac{2B_f}{k}$ at most $k$ times for any $k \in \mathbb{N}$. Further, $f(\boldsymbol{x}_t) \geq f_t^+$ for $\forall t \in \mathbb{N}$. Choose $k = [t/2]$, where $[x]$ is the largest integer smaller than $x$ so that $2k \leq t \leq 2(k+1)$. Then, there exists $k \leq t_k \leq 2k$ so that $f_{t_k}^+ - f_{t_k+1}^+ < \frac{2B_f}{k}$ and $f_{t_k+1}^+ - f(\boldsymbol{x}_{t_k+1}) \leq 0$.

From Lemma B.4,

$$
\begin{aligned}
r_t = f_t^+ - f(\boldsymbol{x}^*) &\leq f_{t_k}^+ - f(\boldsymbol{x}^*) \leq I_{t_k}^f(\boldsymbol{x}^*) \\
&\leq \frac{\tau(B_f)}{\tau(-B_f)} EI_{t_k}^f(\boldsymbol{x}^*) = c_{\tau B} \frac{P_{t_k}(\boldsymbol{x}^*)}{P_{t_k}(\boldsymbol{x}^*)} EI_{t_k}^f(\boldsymbol{x}^*) \\
&\leq c_{\tau B} \frac{P_{t_k}(\boldsymbol{x}_{t_k+1})}{P_{t_k}(\boldsymbol{x}^*)} EI_{t_k}^f(\boldsymbol{x}_{t_k+1}),
\end{aligned}
\tag{42}
$$

where $c_{\tau B} = \frac{\tau(B_f)}{\tau(-B_f)}$. Using $P_t(\boldsymbol{x}) \leq 1$, (42) implies

$$
\begin{aligned}
r_t &\leq \frac{c_{\tau B}}{P_{t_k}(\boldsymbol{x}^*)} \left[ (f_{t_k}^+ - \mu_{t_k}^f(\boldsymbol{x}_{t_k+1}))\Phi(z_{t_k}^f(\boldsymbol{x}_{t_k+1})) + \sigma_{t_k}^f(\boldsymbol{x}_{t_k+1})\phi(z_{t_k}^f(\boldsymbol{x}_{t_k+1})) \right] \\
&\leq \frac{c_{\tau B}}{P_{t_k}(\boldsymbol{x}^*)} \left[ (f_{t_k}^+ - \mu_{t_k}^f(\boldsymbol{x}_{t_k+1}))\Phi(z_{t_k}^f(\boldsymbol{x}_{t_k+1})) + 0.4\sigma_{t_k}^f(\boldsymbol{x}_{t_k+1}) \right],
\end{aligned}
\tag{43}
$$

where the last inequality uses $\phi(\cdot) < 0.4$. From Lemma B.2,

$$
\begin{aligned}
f_{t_k}^+ - \mu_{t_k}^f(\boldsymbol{x}_{t_k+1}) &= f_{t_k}^+ - f_{t_k+1}^+ + f_{t_k+1}^+ - f(\boldsymbol{x}_{t_k+1}) + f(\boldsymbol{x}_{t_k+1}) - \mu_{t_k}^f(\boldsymbol{x}_{t_k+1}) \\
&\leq f_{t_k}^+ - f_{t_k+1}^+ + B_f\sigma_{t_k}^f(\boldsymbol{x}_{t_k+1}) \leq \frac{2B_f}{k} + B_f\sigma_{t_k}^f(\boldsymbol{x}_{t_k+1}).
\end{aligned}
\tag{44}
$$

Using (44) in (43), we have

$$r_t \leq \frac{c_{\tau B}}{P_{t_k}(\boldsymbol{x}^*)} \left[ \frac{2B_f}{k} + (B_f + 0.4)\sigma_{t_k}^f(\boldsymbol{x}_{t_k+1}) \right]. \tag{45}$$

Next, we consider the function $P_{t_k}$ at $\boldsymbol{x}^*$. Using the fact that $c(\boldsymbol{x}^*) \leq 0$, we have by Lemma B.2,

$$\mu_{t_k}^c(\boldsymbol{x}^*) \leq B_c\sigma_{t_k}^c(\boldsymbol{x}^*) + c(\boldsymbol{x}^*) \leq B_c\sigma_{t_k}^c(\boldsymbol{x}^*). \tag{46}$$

Thus,

$$\frac{-\mu_{t_k}^c(\boldsymbol{x}^*)}{\sigma_{t_k}^c(\boldsymbol{x}^*)} \geq -B_c. \tag{47}$$

From the monotonicity of $\Phi$, we have

$$P_{t_k}(\boldsymbol{x}^*) = \Phi\left( \frac{-\mu_{t_k}^c(\boldsymbol{x}^*)}{\sigma_{t_k}^c(\boldsymbol{x}^*)} \right) \geq \Phi(-B_c). \tag{48}$$

Applying (48) to (43), we have

$$r_t \leq \frac{c_{\tau B}}{\Phi(-B_c)} \left[ \frac{2B_f}{k} + (B_f + 0.4)\sigma_{t_k}^f(\boldsymbol{x}_{t_k+1}) \right]. \tag{49}$$

As $t \to \infty$, $t_k \to \infty$ and $k \to \infty$. Further, if $\sigma_t^f(\boldsymbol{x}_{t+1}) \to 0$, $\sigma_{t_k}^f(\boldsymbol{x}_{t_k+1}) \to 0$ and $r_t \to 0$. $\qquad\square$

Proof of Corollary 3.6 is presented next.

*Proof.* We consider the convergence rate for $r_t$ under additional assumptions for the kernel. From Lemma 3.5, for both SE and Matérn kernels, $\sigma_i^f(\boldsymbol{x}_{i+1}) \geq C'k^{-\frac{\min\{\nu,1\}}{d}}\log^\eta(k)$ at most $k$ times for any $k \in \mathbb{N}$ and $i = 1, \ldots, t$.

Choose $k = [t/3]$ so that $3k \leq t \leq 3(k+1)$. Following the proof of Theorem 3.3, there exists $k \leq t_k \leq 3k$ where $f_{t_k}^+ - f_{t_k+1}^+ < \frac{2B_f}{k}$, $f(\boldsymbol{x}_{t_k+1}) \geq f_{t_k+1}^+$, and $\sigma_{t_k}^f(\boldsymbol{x}_{t_k+1}) < C'k^{-\frac{\min\{\nu,1\}}{d}}\log^\eta(k)$. Similar to (49), we can obtain

$$r_t \leq \frac{c_{B_f}}{\Phi(-B_c)} \left[ \frac{2B_f}{k} + (B_f + 0.4)C'k^{-\frac{\min\{\nu,1\}}{d}}\log^\eta(k) \right]. \tag{50}$$

The convergence rates follow. $\qquad\square$

## B.1 Proofs for improved simple regret upper bound under frequentist assumptions

Proof of Theorem 3.7 is given below.

*Proof.* From the proof of Lemma A.5, we know $\sigma_i^f(\boldsymbol{x}_{i+1}) \geq \frac{\sqrt{t\gamma_t^f}}{k}$ at most $k$ times for any $k \leq t$ and $i = 1, \ldots, t$.

Choose $k = [t/3]$ so that $3k \leq t \leq 3(k+1)$. Following the proof of Theorem 3.3, there exists $k \leq t_k \leq 3k$ where $f_{t_k}^+ - f_{t_k+1}^+ < \frac{2B_f}{k}$, $f(\boldsymbol{x}_{t_k+1}) \geq f_{t_k+1}^+$, and $\sigma_{t_k}^f(\boldsymbol{x}_{t_k+1}) < 3\frac{\sqrt{t\gamma_t^f}}{t-3}$. Similar to (50), we can obtain

$$r_t \leq \frac{c_{B_f}}{\Phi(-B_c)} \left[ 3\frac{2B_f}{t-3} + 3(B_f + 0.4)\frac{\sqrt{t\gamma_t^f}}{t-3} \right]. \tag{51}$$

The convergence rates of the simple regret upper bound follow from Lemma A.5. □

We provide the sample complexity of Theorem 3.7 below.

**Corollary B.5.** *Under Assumption 3.2, the CEI algorithm achieves a $\epsilon$ sample complexity of*

$$\mathcal{O}\left( \frac{1}{\epsilon^2}[\log(1/\epsilon)]^{d+1} \right) \text{ and } \mathcal{O}\left( \epsilon^{-\frac{2\nu+d}{\nu}} \log(1/\epsilon) \right), \tag{52}$$

*for SE and Matérn kernels, respectively.*

*Proof.* Using Theorem 3.7, to achieve simple regret of most $\epsilon$ for SE kernel, set

$$t^{-\frac{1}{2}} \log^{\frac{d+1}{2}}(t) = \epsilon.$$

Solving asymptotically for $t$ gives the sample complexity

$$t(\epsilon) = \mathcal{O}\left( \frac{1}{\epsilon^2}[\log(1/\epsilon)]^{d+1} \right).$$

Similarly, for Matérn kernels, we have

$$\epsilon = t^{\frac{-\nu}{2\nu+d}} \log^{\frac{\nu}{2\nu+d}}(t), \tag{53}$$

which completes the proof. □

## C  Proofs for simple regret upper bound under Bayesian objective assumption

We state the boundedness of $f$ and $c$ as a Lemma for easy reference.

**Lemma C.1.** *Under Assumption 3.9, there exists $M_f > 0$ such that $|f(\boldsymbol{x})| \leq M_f$ with probability $\geq 1 - \delta/3$. The constraint function is bounded by its RKHS norm bound $|c(\boldsymbol{x})| \leq B_c$.*

We recall a well-known result on confidence interval of $|f(\boldsymbol{x}) - \mu_t^f(\boldsymbol{x})|$ under Assumption 3.9 [27].

**Lemma C.2.** *Given $\delta \in (0, 1)$, let $\beta = 2\log(1/\delta)$. For any given $\boldsymbol{x} \in C$ and $t \in \mathbb{N}$,*

$$|f(\boldsymbol{x}) - \mu_t^f(\boldsymbol{x})| \leq \sqrt{\beta}\sigma_t^f(\boldsymbol{x}), \tag{54}$$

*holds with probability $\geq 1 - \delta$.*

*Proof.* We prove the inequalities for $f$. Under Assumption 3.9, $f(\boldsymbol{x}) \sim \mathcal{N}(\mu_t(\boldsymbol{x}), \sigma_t^2(\boldsymbol{x}))$. By Lemma A.6,

$$\mathbb{P}\left\{ f(\boldsymbol{x}) - \mu_t^f(\boldsymbol{x}) > \sqrt{\beta}\sigma_t^f(\boldsymbol{x}) \right\} = 1 - \Phi\left( \sqrt{\beta} \right) \leq \frac{1}{2}e^{-\frac{\beta}{2}}. \tag{55}$$

Similarly,

$$\mathbb{P}\left\{ f(\boldsymbol{x}) - \mu_t^f(\boldsymbol{x}) < -\sqrt{\beta}\sigma_t^f(\boldsymbol{x}) \right\} \leq \frac{1}{2}e^{-\frac{\beta}{2}}. \tag{56}$$

Thus,

$$\mathbb{P}\left\{ |f(\boldsymbol{x}) - \mu_t^f(\boldsymbol{x})| < \sqrt{\beta}\sigma_t^f(\boldsymbol{x}) \right\} \geq 1 - e^{-\frac{\beta}{2}}. \tag{57}$$

Let $e^{-\frac{\beta}{2}} = \delta$ and (54) is proven. □

**Lemma C.3.** *Given $\delta \in (0,1)$, let $\beta = 2\log(\pi_t/\delta)$, where $\pi_t = \frac{\pi^2 t^2}{6}$. Then, for all $t \in \mathbb{N}$,*

$$|f(\boldsymbol{x}) - \mu_t^f(\boldsymbol{x})| \leq \sqrt{\beta_t}\sigma_t^f(\boldsymbol{x}), \tag{58}$$

*holds with probability $\geq 1 - \delta$.*

The next lemma address $I_t^f$ under the Bayesian assumption.

**Lemma C.4.** *Under Assumption 3.9, the probability distribution of $I_t^f$ satisfies*

$$\mathbb{P}\{I_t^f(\boldsymbol{x}) \leq a\} = \begin{cases} 0, & a < 0, \\ \Phi\left(\frac{a}{\sigma_t^f(\boldsymbol{x})} - z_t^f(\boldsymbol{x})\right), & a \geq 0. \end{cases} \tag{59}$$

*Proof.* Under Assumption 3.9, at a given $t$, $f(\boldsymbol{x}) \sim \mathcal{N}(\mu_t^f(\boldsymbol{x}), \sigma_t^f(\boldsymbol{x}))$. Since $I_t^f(\boldsymbol{x}) \geq 0$ for all $\boldsymbol{x}$, (59) follows immediately if $a < 0$. For $a \geq 0$,

$$\mathbb{P}\{I_t^f(\boldsymbol{x}) \leq a\} = \mathbb{P}\{f_t^+ - f(\boldsymbol{x}) \leq a\} = 1 - \mathbb{P}\{f(\boldsymbol{x}) \leq f_t^+ - a\}.$$

Using basic properties of the standard normal CDF,

$$1 - \mathbb{P}\{f(\boldsymbol{x}) \leq f_t^+ - a\} = 1 - \Phi\left(\frac{f_t^+ - a - \mu_t^f(\boldsymbol{x})}{\sigma_t^f(\boldsymbol{x})}\right) = \Phi\left(\frac{a - f_t^+ + \mu_t^f(\boldsymbol{x})}{\sigma_t^f(\boldsymbol{x})}\right).$$

$\square$

Next, we present the relationship between $I_t^f(\boldsymbol{x})$ and $EI_t^f(\boldsymbol{x})$.

**Lemma C.5.** *Given $\delta \in (0,1)$, let $\beta = \max\{1.44, 2\log(c_\alpha/\delta)\}$, where constant $c_\alpha = \frac{1+2\pi}{2\pi}$. Under Assumption 3.9, at given $\boldsymbol{x} \in C$ and $t \in \mathbb{N}$,*

$$\mathbb{P}\left\{|I_t^f(\boldsymbol{x}) - EI_t^f(\boldsymbol{x})| \leq \sqrt{\beta}\sigma_t^f(\boldsymbol{x})\right\} \geq 1 - \delta. \tag{60}$$

*Proof.* Given a scalar $w > 1$, we consider the probabilities

$$\mathbb{P}\left\{I_t^f(\boldsymbol{x}) > \sigma_t^f(\boldsymbol{x})w + EI_t^f(\boldsymbol{x})\right\} \quad \text{and} \quad \mathbb{P}\left\{I_t(\boldsymbol{x}) < -\sigma_t^f(\boldsymbol{x})w + EI_t^f(\boldsymbol{x})\right\}. \tag{61}$$

Consider the first probability in (61). From Lemma A.7, $EI_t(\boldsymbol{x}) \geq 0$ for $\forall \boldsymbol{x}$ and $t$. Therefore, $\sigma_t(\boldsymbol{x})w + EI_t(\boldsymbol{x}) > 0$. From Lemma A.7, Lemma C.4, and the monotonicity of $\Phi$, we have

$$\mathbb{P}\left\{I_t^f(\boldsymbol{x}) > \sigma_t^f(\boldsymbol{x})w + EI_t^f(\boldsymbol{x})\right\} = 1 - \Phi\left(\frac{\sigma_t^f(\boldsymbol{x})w + EI_t^f(\boldsymbol{x}) - f_t^+ + \mu_t^f(\boldsymbol{x})}{\sigma_t^f(\boldsymbol{x})}\right)$$

$$\leq 1 - \Phi(w) \leq \frac{1}{2}e^{-\frac{w^2}{2}}, \tag{62}$$

where the last inequality is from Lemma A.6.

For the second probability in (61), we further distinguish between two cases. First, consider $-\sigma_t^f(\boldsymbol{x})w + EI_t^f(\boldsymbol{x}) < 0$. From Lemma C.4,

$$\mathbb{P}\left\{I_t^f(\boldsymbol{x}) < -\sigma_t^f(\boldsymbol{x})w + EI_t(\boldsymbol{x})\right\} = 0. \tag{63}$$

Second, consider the premise $-\sigma_t^f(\boldsymbol{x})w + EI_t^f(\boldsymbol{x}) \geq 0$. By Lemma C.4, we have

$$\mathbb{P}\left\{I_t^f(\boldsymbol{x}) < -\sigma_t^f(\boldsymbol{x})w + EI_t^f(\boldsymbol{x})\right\} = \Phi\left(-w + \frac{EI_t^f(\boldsymbol{x}) - f_t^+ + \mu_t^f(\boldsymbol{x})}{\sigma_t^f(\boldsymbol{x})}\right). \tag{64}$$

To proceed, we show that $f_t^+ - \mu_t^f(\boldsymbol{x}) \geq 0$. Suppose on the contrary, $f_t^+ - \mu_t^f(\boldsymbol{x}) < 0$ and thus $z_t^f(\boldsymbol{x}) < 0$. From Lemma A.7,

$$\frac{EI_t^f(\boldsymbol{x})}{\sigma_t^f(\boldsymbol{x})} < \phi(z_t^f(\boldsymbol{x})) \leq \phi(0) < 1 \leq w, \tag{65}$$

which contradicts the premise of this case. Thus, we have $f_t^+ - \mu_t^f(\boldsymbol{x}) \geq 0$ (and $z_t^f(\boldsymbol{x}) \geq 0$). From the definition (4), since $\Phi \in (0, 1)$,

$$\frac{EI_t^f(\boldsymbol{x}) - f_t^+ + \mu_t^f(\boldsymbol{x})}{\sigma_t^f(\boldsymbol{x})} = \left[ z_t^f(\boldsymbol{x}) \left( \Phi(z_t^f(\boldsymbol{x})) - 1 \right) + \phi(z_t^f(\boldsymbol{x})) \right] < \phi(z_t^f(\boldsymbol{x})). \tag{66}$$

In addition, by the premise of this case and Lemma A.7,

$$w \leq \frac{EI_t^f(\boldsymbol{x})}{\sigma_t^f(\boldsymbol{x})} \leq z_t^f(\boldsymbol{x}) + \phi(z_t^f(\boldsymbol{x})). \tag{67}$$

Given that $w > 1$ and $\phi(0) \geq \phi(z_t^f(\boldsymbol{x}))$, we have

$$z_t^f(\boldsymbol{x}) + \phi(0) > z_t^f(\boldsymbol{x}) + \phi(z_t^f(\boldsymbol{x})) > w, \ z_t^f(\boldsymbol{x}) > w - \phi(0) > 0. \tag{68}$$

As $z_t(\boldsymbol{x}) \geq 0$ increases, $\phi(z_t(\boldsymbol{x})) > 0$ decreases. Thus, we have

$$\frac{z_t^f(\boldsymbol{x})}{\phi(z_t^f(\boldsymbol{x}))} > \frac{w - \phi(0)}{\phi(w - \phi(0))}, \ \phi(z_t^f(\boldsymbol{x})) < \frac{\phi(w - \phi(0))}{w - \phi(0)} z_t^f(\boldsymbol{x}). \tag{69}$$

Denote $c_1(w) = \frac{w - \phi(0)}{w - \phi(0) + \phi(w - \phi(0))}$. Applying (69) to (67), we obtain

$$c_1(w)w < z_t^f(\boldsymbol{x}), \ \ \phi(z_t^f(\boldsymbol{x})) < \phi(c_1(w)w). \tag{70}$$

Applying (70) and (66) to (64), we obtain

$$\mathbb{P}\left\{ I_t^f(\boldsymbol{x}) < -w\sigma_t^f(\boldsymbol{x}) + EI_t^f(\boldsymbol{x}) \right\} < \Phi\left( -w + \phi(z_t^f(\boldsymbol{x})) \right)$$
$$< \Phi\left( -w + \phi(c_1(w)w) \right). \tag{71}$$

Notice that $\phi(c_1(w)w) < \phi(c_1(w)) < \phi(c_1(w))w$ due to $w > 1$. By the definition of $\Phi$ and mean value theorem,

$$\Phi\left( -w + \phi(c_1(w)w) \right) = \Phi(-w) + \int_{-w}^{-w + \phi(c_1(w)w)} \frac{1}{\sqrt{2\pi}} e^{-\frac{1}{2}x^2} dx \leq \Phi(-w) +$$
$$\frac{1}{\sqrt{2\pi}} e^{-\frac{1}{2}(w - \phi(c_1(w)w))^2} \phi(c_1(w)w) \leq \Phi(-w) + \frac{1}{2\pi} e^{-\frac{1}{2}((1 - \phi(c_1(w)))w)^2} e^{-\frac{1}{2}(c_1(w)w)^2} \tag{72}$$
$$\leq \Phi(-w) + \frac{1}{2\pi} e^{-\frac{1}{2}c_2(w)w^2} \leq \frac{1}{2} e^{-\frac{1}{2}w^2} + \frac{1}{2\pi} e^{-\frac{1}{2}c_2(w)w^2},$$

where $c_2(w) = [1 - \phi(c_1(w))]^2 + [c_1(w)]^2$. The last inequality in (72) again uses Lemma A.6. Notice that $c_2(w)$ increases with $w$ and for $w \geq 1.2$, $c_2(w) > 1$. Thus, $e^{-\frac{1}{2}w^2} > e^{-\frac{1}{2}c_2(w)w^2}$ for $w \geq 1.2$, which simplifies (72) to

$$\Phi\left( -w + \phi(c_1(w)w) \right) < c_{\pi 1} e^{-\frac{1}{2}w^2}. \tag{73}$$

where $c_{\pi 1} = \frac{1 + \pi}{2\pi}$. Therefore, by (71) and (73), if $w \geq 1.2$,

$$\mathbb{P}\left\{ I_t(\boldsymbol{x}) < -\sigma_t(\boldsymbol{x})w + EI_t(\boldsymbol{x}) \right\} < c_{\pi 1} e^{-\frac{1}{2}w^2}. \tag{74}$$

Combining (74) with (62) and (63), we have

$$\mathbb{P}\left\{ \left| I_t^f(\boldsymbol{x}) - EI_t^f(\boldsymbol{x}) \right| > w\sigma_t^f(\boldsymbol{x}) \right\} < c_\alpha e^{-\frac{1}{2}w^2}, \tag{75}$$

where $c_\alpha = \frac{1 + 2\pi}{2\pi}$ for $w \geq 1.2$. The probability in (75) monotonically decreases with $w$. Let $\delta = c_\alpha e^{-\frac{1}{2}w^2}$. Then, taking the logarithm of $\delta$ leads to $\log(\frac{1 + 2\pi}{2\pi\delta}) = \frac{1}{2}w^2$. Let $\beta = \max\{w^2, 1.2^2\}$, and the proof is complete. $\square$

The relationship between $I_t^f(\boldsymbol{x})$ and $EI_t^f(\boldsymbol{x})$ under the GP prior assumption is given in the following lemma.

**Lemma C.6.** *Given $\delta \in (0,1)$, let $\beta = 2\log(2c_\alpha/\delta)$, where $c_\alpha = \frac{1+2\pi}{2\pi}$. At given $\boldsymbol{x} \in C$ and $t \in \mathbb{N}$,*

$$\frac{\tau(-\sqrt{\beta})}{\tau(\sqrt{\beta})} I_t^f(\boldsymbol{x}) \le EI_t^f(\boldsymbol{x}), \tag{76}$$

*holds with probability $\ge 1 - \delta$*

*Proof.* From Lemma C.2, with probability $\ge 1 - \delta$, (54) stands. If $f_t^+ - f(\boldsymbol{x}) \le 0$, then $I_t(\boldsymbol{x}) = 0$. Since $EI_t(\boldsymbol{x}) \ge 0$, (76) is trivial. If $f_t^+ - f(\boldsymbol{x}) > 0$, by Lemma C.2,

$$\begin{aligned} f_t^+ - \mu_t^f(\boldsymbol{x}) = f_t^+ - f(\boldsymbol{x}) + f(\boldsymbol{x}) - \mu_t^f(\boldsymbol{x}) &> f(\boldsymbol{x}) - \mu_t^f(\boldsymbol{x}) \\ &> -\sqrt{\beta}\sigma_t^f(\boldsymbol{x}), \end{aligned} \tag{77}$$

with probability greater than $\ge 1 - \delta/2$. From the monotonicity of $\tau$, we have

$$\tau\left(\frac{f_t^+ - \mu_t^f(\boldsymbol{x})}{\sigma_t^f(\boldsymbol{x})}\right) > \tau(-\sqrt{\beta}), \tag{78}$$

and therefore,

$$EI_t^f(\boldsymbol{x}) = \sigma_t^f(\boldsymbol{x})\tau\left(\frac{f_t^+ - \mu_t^f(\boldsymbol{x})}{\sigma_t^f(\boldsymbol{x})}\right) > \tau(-\sqrt{\beta})\sigma_t^f(\boldsymbol{x}), \tag{79}$$

with probability greater than $1 - \delta/2$. Using $\delta/2$ in Lemma C.5,

$$I_t^f(\boldsymbol{x}) - EI_t^f(\boldsymbol{x}) \le \sqrt{\beta}\sigma_t^f(\boldsymbol{x}), \tag{80}$$

with probability $\ge 1 - \delta/2$. Applying (80) to (79) with union bound leads to

$$EI_t^f(\boldsymbol{x}) > \frac{\tau(-\sqrt{\beta})}{\sqrt{\beta} + \tau(-\sqrt{\beta})} I_t^f(\boldsymbol{x}) = \frac{\tau(-\sqrt{\beta})}{\tau(\sqrt{\beta})} I_t^f(\boldsymbol{x}), \tag{81}$$

with probability greater than $1 - \delta$. $\qquad\square$

We can now prove Theorem 3.10.

*Proof.* From Lemma C.1,

$$\sum_{i=0}^{t-1} f_i^+ - f_{i+1}^+ = f_0^+ - f_t^+ \le 2M_f, \tag{82}$$

with probability $\ge 1 - \delta/3$. Next, consider $f_i^+ - \mu_i^f(x_{i+1})$. Recall that $\beta_t = 2\log(3\pi_t/\delta)$. From Lemma C.3 Lemma A.4 and Lemma A.3, we have

$$\begin{aligned} \sum_{i=0}^{t-1} \max\{f_i^+ - \mu_i^f(\boldsymbol{x}_{i+1}), 0\} &= \sum_{i=0}^{t-1} \max\{f_i^+ - f(\boldsymbol{x}_{i+1}) + f(\boldsymbol{x}_{i+1}) - \mu_i^f(\boldsymbol{x}_{i+1}), 0\} \\ &\le \sum_{i=0}^{t-1} f_i^+ - f_{i+1}^+ + \beta_t^{1/2}\sigma_i^f(\boldsymbol{x}_{i+1}) \le 2M_f + \beta_t^{1/2}\sqrt{C_\gamma t\gamma_t^f}, \end{aligned} \tag{83}$$

with probability $\ge 1 - 2\delta/3$ via union bound. Given that $\max\{f_i^+ - \mu_i^f(\boldsymbol{x}_{i+1}), 0\} \ge 0$, $\max\{f_i^+ - \mu_i^f(\boldsymbol{x}_{i+1}), 0\} \ge \frac{2M_f}{k} + \frac{\beta_t^{1/2}}{k}\sqrt{C_\gamma t\gamma_t^f}$ at most $k$ times for any $k \in \mathbb{N}$ with probability $\ge 1 - 2\delta/3$. Choose $k = [t/2]$, where $[x]$ is the largest integer smaller than $x$ so that $2k \le t \le 2(k+1)$. Then, choose the first index $t_k$ where $k \le t_k \le 2k$ so that $\max\{f_{t_k}^+ - \mu_{t_k}^f(\boldsymbol{x}_{t_k+1}), 0\} < \frac{2M_f}{k} + \frac{\beta_t^{1/2}}{k}\sqrt{C_\gamma t\gamma_t^f}$. As discussed above, such a $t_k$ exists with probability $\ge 1 - 2\delta/3$. We note that maximum information gain and its upper bound does not depend on the optimization path. Importantly, the choice of $t_k$ does not depend on random information after iteration $t_k$.

From Lemma C.6 and $\beta = 2\log(6c_\alpha/\delta)$, with probability $\geq 1 - \delta/3$,

$$
\begin{aligned}
r_t =& f_t^+ - f(\boldsymbol{x}^*) \leq f_{t_k}^+ - f(\boldsymbol{x}^*) \leq I_{t_k}^f(\boldsymbol{x}^*) \\
\leq & \frac{\tau(\beta^{1/2})}{\tau(-\beta^{1/2})} EI_{t_k}^f(\boldsymbol{x}^*) = c_\tau(\beta)\frac{P_{t_k}(\boldsymbol{x}^*)}{P_{t_k}(\boldsymbol{x}^*)} EI_{t_k}^f(\boldsymbol{x}^*) \leq c_\tau(\beta)\frac{P_{t_k}(\boldsymbol{x}_{t_k+1})}{P_{t_k}(\boldsymbol{x}^*)} EI_{t_k}^f(\boldsymbol{x}_{t_k+1}) \\
=& c_\tau(\beta)\frac{P_{t_k}(\boldsymbol{x}_{t_k+1})}{P_{t_k}(\boldsymbol{x}^*)}\left[ (f_{t_k}^+ - \mu_{t_k}^f(\boldsymbol{x}_{t_k+1}))\Phi(z_{t_k}^f(\boldsymbol{x}_{t_k+1})) + \sigma_{t_k}^f(\boldsymbol{x}_{t_k+1})\phi(z_{t_k}^f(\boldsymbol{x}_{t_k+1})) \right] \\
\leq& c_\tau(\beta)\frac{P_{t_k}(\boldsymbol{x}_{t_k+1})}{P_{t_k}(\boldsymbol{x}^*)}\left[ (f_{t_k}^+ - \mu_{t_k}^f(\boldsymbol{x}_{t_k+1}))\Phi(z_{t_k}^f(\boldsymbol{x}_{t_k+1})) + 0.4\sigma_{t_k}^f(\boldsymbol{x}_{t_k+1}) \right],
\end{aligned}
\tag{84}
$$

where the last inequality uses $\phi(\cdot) < 0.4$. From the choice of $t_k$, (84) leads to

$$
r_t \leq c_\tau(\beta)\frac{P_{t_k}(\boldsymbol{x}_{t_k+1})}{P_{t_k}(\boldsymbol{x}^*)}\left[ \frac{2M_f}{k} + \frac{\beta_t^{1/2}}{k}\sqrt{C_\gamma t\gamma_t^f} + (0.4 + \beta^{1/2})\sigma_{t_k}^f(\boldsymbol{x}_{t_k+1}) \right],
\tag{85}
$$

with probability $\geq 1 - \delta$. Next, we consider the probability function $P_{t_k}$ at $\boldsymbol{x}^*$ and $\boldsymbol{x}_{t_k+1}$. Using the fact that $c(\boldsymbol{x}^*) \leq 0$, we have by Lemma C.2 at $\boldsymbol{x}^*$ and $t_k$,

$$
\mu_{t_k}^c(\boldsymbol{x}^*) \leq B_c\sigma_{t_k}^c(\boldsymbol{x}^*) + c(\boldsymbol{x}^*) \leq B_c\sigma_{t_k}^c(\boldsymbol{x}^*).
\tag{86}
$$

Thus, we can write

$$
\frac{-\mu_{t_k}^c(\boldsymbol{x}^*)}{\sigma_{t_k}^c(\boldsymbol{x}^*)} \geq -B_c.
\tag{87}
$$

From the monotonicity of $\Phi$, we have

$$
\Phi\left( \frac{-\mu_{t_k}^c(\boldsymbol{x}^*)}{\sigma_{t_k}^c(\boldsymbol{x}^*)} \right) \geq \Phi\left( -B_c \right),
\tag{88}
$$

Using (88), the $P_{t_k}$ functions have

$$
\frac{P_{t_k}(\boldsymbol{x}_{t_k+1})}{P_{t_k}(\boldsymbol{x}^*)} \leq \frac{1}{\Phi(-B_c)}.
\tag{89}
$$

Applying (89) to (85), we have

$$
r_t \leq c_\tau(\beta)\frac{1}{\Phi(-B_c)}\left[ \frac{2M_f}{k} + \frac{\beta_t^{1/2}}{k}\sqrt{C_\gamma t\gamma_t^f} + (0.4 + \beta^{1/2})\sigma_{t_k}^f(\boldsymbol{x}_{t_k+1}) \right],
\tag{90}
$$

with probability $\geq 1 - \delta$.

$\square$

The proof of Theorem 3.11 is next.

*Proof.* From Lemma A.5, $\sigma_i^f(\boldsymbol{x}_{i+1}) \geq \frac{\sqrt{\gamma_t^f t}}{k}$, where $i = 0, \ldots, t-1$, at most $k$ times for any $k \in \mathbb{N}$ and $k \leq t$. Choose $k = \lceil t/3 \rceil$ which leads to $3k \leq t \leq 3(k+1)$. Let $t_k$ be the first index in $[k, 3k]$ so that $\sigma_{t_k}^f(\boldsymbol{x}_{t_k+1}) \leq \frac{\sqrt{t\gamma_t^f}}{k}$ and $\max\{f_{t_k}^+ - \mu_{t_k}^f(\boldsymbol{x}_{t_k+1}), 0\} < \frac{2M_f}{k} + \frac{\beta_t^{1/2}}{k}\sqrt{C_\gamma t\gamma_t^f}$, which exists with probablity $\geq 1 - 2\delta/3$. Notice that $\beta_t^{1/2} = \mathcal{O}(\log^{1/2}(t))$. Following the proof for (90), we have

$$
r_t \leq c_\tau(\beta)\frac{1}{\Phi(-B_c)}\left[ \frac{2M_f}{k} + \frac{\beta_t^{1/2}}{k}\sqrt{C_\gamma t\gamma_t^f} + (0.4 + \beta^{1/2})\frac{\sqrt{t\gamma_t^f}}{k} \right],
\tag{91}
$$

with probability $\geq 1 - \delta$. Using Lemma A.2, the proof is complete.

$\square$

# D    Test problems

The mathematical formulations of the testing problems in Section 4.2 are given in this section. The objective, constraint functions and the optimal $f$ of Problem 1 is given below.

$$
\begin{aligned}
f(\boldsymbol{x}) &= \sin(x_1) + x_2, \\
c(\boldsymbol{x}) &= \sin(x_1)\sin(x_2) + 0.95 \leq 0, \\
x_i &\in [0,6], i = 1,2, \\
f^* &= 0.25.
\end{aligned} \tag{92}
$$

The objective, constraint functions and the optimal $f$ of Problem 2 is given below.

$$
\begin{aligned}
f(\mathbf{x}) &= x_1 + x_2 \\
c_1(\boldsymbol{x}) &= -0.5\sin(2\pi(x_1^2 - 2x_2)) - x_1 - 2x_2 + 1.5 \leq 0 \\
c_2(\boldsymbol{x}) &= x_1^2 + x_2^2 - 1.5 \leq 0 \\
x_i &\in [0,1], i = 1,2 \\
f^* &= 0.6.
\end{aligned} \tag{93}
$$

The objective, constraint functions and the optimal $f$ of Problem 3 is given below.

$$
\begin{aligned}
f(\mathbf{x}) &= x_1 + x_2 + x_3 + x_4 \\
c_1 &= 1.1 - \sum_{i=1}^{4} E_i \exp\left(\sum_{j=1}^{4} -A_{j,i}(x_j - P_{j,i})^2\right) \\
x_i &\in [0,1], i = 1, \ldots, 4 \\
E &= [1, 1.2, 3, 3.2]^\top \\
P &= \begin{bmatrix} 0.131 & 0.232 & 0.234 & 0.404 \\ 0.169 & 0.413 & 0.145 & 0.882 \\ 0.556 & 0.830 & 0.352 & 0.873 \\ 0.012 & 0.373 & 0.288 & 0.574 \end{bmatrix} \\
A &= \begin{bmatrix} 10 & 0.05 & 3 & 17 \\ 3 & 10 & 3.5 & 8 \\ 17 & 17 & 1.7 & 0.05 \\ 3.5 & 0.1 & 10 & 10 \end{bmatrix} \\
f^* &= 0.
\end{aligned} \tag{94}
$$

The objective, constraint functions and the optimal $f$ of Problem 4 is given below.

$$
\begin{aligned}
f(\mathbf{x}) &= -\sum_{i=1}^{4} \alpha_i \exp\left(-\sum_{j=1}^{6} A_{ij}(x_j - P_{ij})^2\right) \\
c(\boldsymbol{x}) &= \sum_{j=1}^{4} x_j - 3 \\
x_i &\in [0,1], i = 1, \ldots, 6 \\
\alpha &= [1.0, 1.2, 3.0, 3.2]^\top \\
A &= \begin{bmatrix} 10 & 3.0 & 17 & 3.5 & 1.7 & 8.0 \\ 0.05 & 10 & 17 & 0.1 & 8.0 & 14 \\ 3.0 & 3.5 & 1.7 & 10 & 17 & 8.0 \\ 17 & 8.0 & 0.05 & 10 & 0.1 & 14 \end{bmatrix} \\
P &= \begin{bmatrix} 0.131 & 0.170 & 0.557 & 0.012 & 0.828 & 0.587 \\ 0.233 & 0.414 & 0.831 & 0.374 & 0.100 & 0.999 \\ 0.235 & 0.145 & 0.352 & 0.288 & 0.305 & 0.665 \\ 0.405 & 0.883 & 0.873 & 0.574 & 0.109 & 0.038 \end{bmatrix} \\
f^* &= -3.32.
\end{aligned} \tag{95}
$$

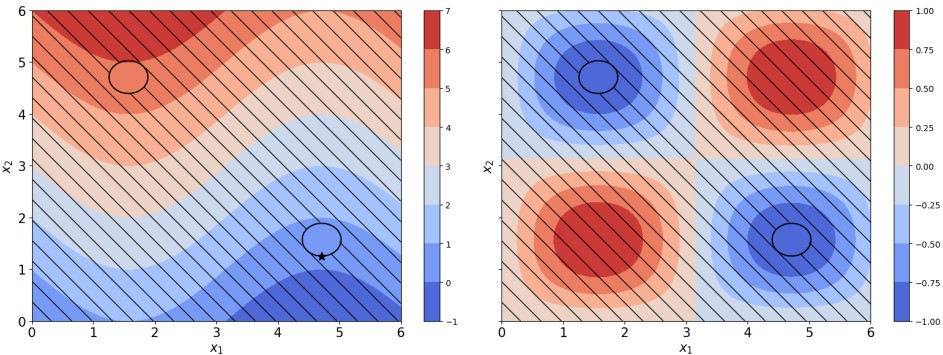

Figure 3: Contour plots for the objective function (left) and constraint function (right) for Problem 1. The infeasible region is marked on the plots. The global optimum is marked with a star sign.

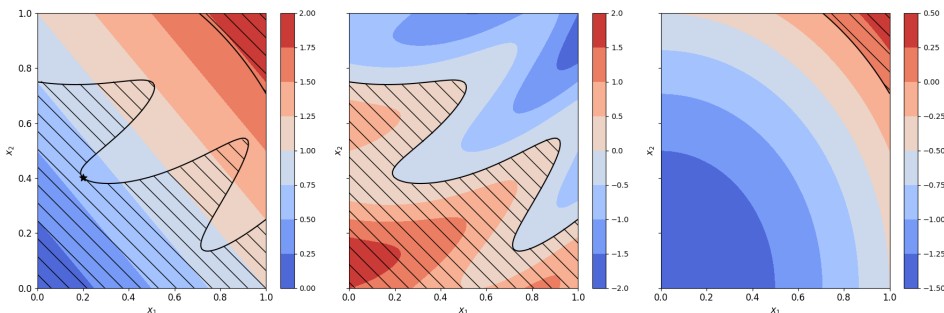

Figure 4: Contour plots for the objective function (left) and the two constraint functions (middle and right) for Problem 2. The infeasible region is marked with black line on the objective contour. The global optimum is marked with a star sign.

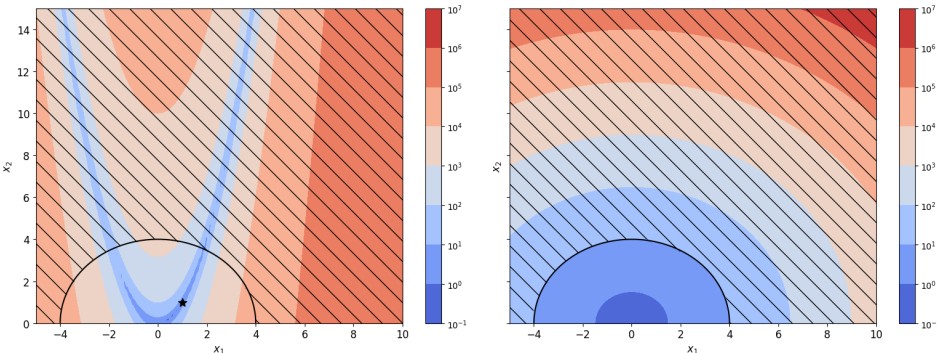

Figure 5: Contour plots for the objective function (left) and constraint function (right) for Problem 5. The infeasible region is marked on the plots. The global optimum is marked with a star sign.

The objective, constraint functions and the optimal $f$ of Problem 5 is given below.

$$
\begin{aligned}
f(\mathbf{x}) &= 100(x_2 - x_1^2)^2 + (1 - x_1)^2 \\
c_1(\boldsymbol{x}) &= \sqrt{x_1^2 + x_1^2} - 4 \\
c_2(\boldsymbol{x}) &= x_1^2 + x_2^2 - 1.5, \\
x_1 &\in [-5, 10], \ x_2 \in [0, 15] \\
f^* &= 0.
\end{aligned}
\tag{96}
$$

The contour plots of Problem 1, 2, and 5 are given in Figure 3, 4, and 5.

