# OpenReview forum: "Convergence Rates of Constrained Expected Improvement"
_NeurIPS.cc/2025/Conference — NeurIPS 2025 spotlight_

### Official Review · Reviewer_3QQu · 2025-07-02

**Clarity:** 2
**Significance:** 3
**Originality:** 4
**Rating:** 5
**Confidence:** 3

**Summary:**

This paper considers the constrained Bayesian optimization framework where one desires to minimize the function $f(x)$ is subject to the constraint $c(x) \leq 0$, where both are black box functions. The expected improvement algorithm, based on a Gaussian process model, is a celebrated algorithm in the unconstrained setting. The constrained expected improvement (CEI) algorithm extends this by including a term in the expected improvement that accounts for the feasibility $c(x) \leq 0$. This paper presents, to my knowledge, first convergence guarantees or regret bounds for the CEI algorithm. Pacifically, in both a frequency setting where are the functions are assumed to come from an RKHS, and also an amazing setting where they are assumes sampled from the Gaussen process, the authors prove appropriate convergence rates of the regret, which extends the results in the unconstrained case. The authors also specialize these results to the commonly used squared exponential and Matérn kernels, and present some numerics to show convergence of the regret.

**Questions:**

1. Please see my question/comment on Weakness 1 above.
2. Please see my question/comment on Weakness 2 above.
3. Starting at the beginning of Section 2.1, there are a few instances of subscript $k_f$. Is this a typo? Should it be $k^f$?
4. Why do you refer to the regret as "simple"?
5. Theorem 3.7 seems more like a Corollary of Theorem 3.4 or is my understanding wrong? I guess Thm 3.4 is a generic result for all kernels, Thm 3.7 is its specialization to SE and Matérn kernels, and Thm 3.8 is a strengthening that uses the special structure of SE and Matérn kernels.
6. Line 217:  Should Theorem 3.8 be Theorem 3.4?
7. In Section 3.2, is the probability $\delta$ over both sampling of $f$ and $c$? Is this probability affected or does it somehow affect the results more when you have more than 1 constraint?

**Ethical Concerns:**

["NO or VERY MINOR ethics concerns only"]

**Final Justification:**

The authors have strongly addressed my major weaknesses from the initial review. In particular, new theoretical bounds that allow for constraint violations are an excellent addition to the paper. As such, I have raised my score from 3 to 5.

**Limitations:**

yes

**Quality:**

2

**Strengths And Weaknesses:**

Assessment:
In my view this is a moderately high quality submission in its current state. The results are technically sound. However, there are perhaps a few gaps in the results that make the paper a bit less mature than ideal (please see my detailed comments/questions below) in its current form. These might be able to be addressed in the rebuttal period. Although I did not read the appendix in detail, the claims appear well supported by extensive proofs. The paper is also written and organized OK; it clearly articulates the main ideas and results but the grammar and style could be improved somewhat throughout.

Strengths:
1. The results of this paper are significant, interesting, and have potential for impact. I like the topic, as I think Bayesian optimization is becoming more and more popular and important, both in theory and practice. Convergence rates for important Bayesian optimization algorithms, like CEI, is therefore fundamental and important.
2. It is very good that this paper provides precise convergence rates for the regret, that all of the constants are explicitly spelled out, and that the arguments are outlined. I quite appreciate these level of transparency in the results.
3. I am not the most well versed in the closely related recent literature. However, to the best of my knowledge, there are no overlapping works in the literature and this work appears to present original new insights, which I think will be valuable.

Weaknesses:
To me the weaknesses concern two aspects of the bounds. These weaknesses are the main reasons for my current values of the Quality and Overall Score below, both of which could possibly be improved in the rebuttal. I list them below in my view on order of importance.

1. I understand your rationale for defining regret as you do, but I wonder if by doing so you are being ``overly strict" in the requirements of feasibility, leading to perhaps overly conservative bounds. What I mean is that, while the CEI expected improvement criteria is somewhat symmetric in how it multiples the constraint and objective criteria, your analysis is not. Instead, you require to only consider those points $x_t$ that actually are feasible with $c(x_t)$. As a result, many points may be thrown away. Furthermore, I think this manifests in the bound in the term $1/\Phi(-B_c)$, which only gets worse as a product of multiple terms with multiple constraints (Remark 3.15). Is it possible to have a more "parameterized" version of a bound that might allow small violations in constraints $c(x_t) \leq \epsilon$ at the benefit of rejecting less points? I can imagine scenarios where decision-makers might be OK with this. And you can also recover your original bound by setting $\epsilon = 0$. I am very curious to know your thoughts and if this approach is feasible, as I think it would improve the bounds a lot.
2. To my understanding, your bounds implicitly involve a well-specified assumption where the kernel in Assumption 3.3/3.10 is the same as the kernel in the GP model in the algorithm; is this correct? If so, at the very least I think this must be acknowledged up front as a limitation of the current analysis. It is also of course interesting to see if such assumptions can be relaxed, but that might be beyond the scope of this current submission.
3. I furthermore have several other questions below, which I don't inherently view as weaknesses per se, but I would appreciate to see addressed.

---

> ### Author Rebuttal · Authors · 2025-07-30
>
> We first want to thank the reviewer for the insightful comments and suggestions. We hope to address these concerns below.
>
> $\textbf{weakness 1}$ $\textit{Strict point selection and constraint tolerance.}$
>
> R: We greatly appreciate the reviewer's suggestion and we are happy to report that we have performed the analysis that includes constraint tolerance $\epsilon$. The simple regret bound is affected by $\epsilon$ and should lead to
>  $1/\Phi(\epsilon-B_c)$ in place of $1/\Phi(-B_c)$, which is an improvement.  The convergence rate remains similar since it is dominated by the maximum information gain of $f$.
>
> Moreover, we can proceed further regarding the term $1/\Phi(\epsilon-B_c)$, which strongly relates to reviewer's thought that the bounds are too conservative. In our analysis, with the inclusion of $\epsilon$, we in fact are able to reach the bound with $1/\Phi(\frac{\epsilon}{\sigma_{t_k}^c(x^*)}-B_c)$.
>
> This comes from the confidence interval of $c$ at $t_k$ and $x^* $ and $c(x^* )\leq 0$.
>
> Recall that $x^* $ is an optimal solution. As CEI gets closer to a solution $x^* $, we should have $\frac{\epsilon}{\sigma_{t_k}^c(x^* )}\to \infty$ for $\epsilon>0$. That is, this coefficient $1/\Phi(\frac{\epsilon}{\sigma_{t_k}^c(x^*)}-B_c)$ is time varying and tends to 1, thereby resolving the reviewer's concern about how large $1/\Phi(-B_c)$ can become. From here, we use $\sigma_{t_k}(x)\leq 1$ to get $1/\Phi(\epsilon-B_c)$.
>
> We can opt to keep  $1/\Phi(\frac{\epsilon}{\sigma_{t_k}^c(x^* )}-B_c)$ in the bound, but $\sigma_{t_k}^c(x^* )$ is uncommon in a regret bound.
> Ideally, we can write an upper bound for $\sigma_{t_k}^c(x^* )$ and use it in the simple regret bound. However, providing a bound or a rate for $\sigma_{t_k}^c(x^* )$ is very challenging. To our best knowledge, such an upper bound for $\sigma_{t_k} (x^* )$ in unconstrained EI is not available. Therefore, we use the rather relaxed bound $\sigma_{t_k}^c(x^* ) \leq 1$ and $1/\Phi(\epsilon-B_c)$. We have added a remark how this term in fact decreases with $t$. An upper bound bound on $\sigma_{t_k}^c(x^* )$ is a technical challenge we can purse in the future.
>
> We have added the newly derived results to the revised paper. Indeed, a constraint tolerance is standard in both constrained BO algorithms [1] and general gradient-based optimization, $\textit{e.g.}$, Ipopt [2].
> We believe this added tolerance could also help make the CEI algorithm more robust when there is small noise in the observation of the constraints.
>
> $\textbf{weakness 2}$ $\textit{Smoothness assumption.}$
>
> R:Yes, the kernel in Assumption 3.3/3.10 is the same as the kernel in the GP model in the algorithm. We acknowledge this as a limitation of our analysis and have clarified it more explicitly in the paper. To the best of our knowledge, this is a standard assumption in the existing literature.
>
> Relaxing these assumptions has been a topic of interest to us, where relevant papers include [3].  To maintain the confidence intervals between the GP models and the functions $f$ and $c$, we think some kind of quantification of distance is needed between the RKHS spaces of these functions and the models. One premature thought we have had is to connect the smoothness assumptions of $f$ with the common assumptions used in the gradient-based optimization literature. For instance, growth conditions have led to improvement of regret bounds [4].
>
> $\textbf{Question 3}$ $\textit{A typo.}$
>
> R: Yes. Thank you for pointing out the typos and we have fixed them all.
>
> $\textbf{Question 4}$ $\textit{Simple regret name.}$
>
> R: We called it simple because in the unconstrained case where every sample is feasible, the regret would reduce to the standard simple regret that is widely adopted in the BO literature.
>
> $\textbf{Question 5}$ $\textit{Theorem should be corollary.}$
>
> R:Yes the reviewer is correct. We use theorem for both 3.4 and 3.7 because 3.4 is an upper bound and 3.7 is the convergence rate. Both can be considered theorem-level results and they are often presented together in one theorem. However, in our case, we deliver two different rates in Theorem 3.7 and 3.8 using different techniques. That is the reason why 3.4 is separated from 3.7 and 3.8 for clarity of presentation. Following your suggestion, we have changed Theorem 3.7 to a Corollary.
>
>
> $\textbf{Question 6}$ $\textit{Another typo.}$
>
> R: Yes the reviewer is correct. We have corrected it.
>
> $\textbf{Question 7}$ $\textit{Probability of multiple constraints.}$
>
> R: Yes it is the probability concerning both $f$ and $c$. And yes increasing the number of constraints would change the constant $6$ in $\beta$. For instance, if we have $2$ constraints and want to maintain equal probability of confidence intervals on $f$, $c_1$ and $c_2$, then we should set $\beta=2\log(8c_{\alpha}/\delta)$. Consequently, this would not change the convergence rate.
>
> [1] S. Ariafar, J. Coll-Font, D. Brooks, and J. Dy. ADMMBO: Bayesian optimization with unknown
> constraints using ADMM. Journal of Machine Learning Research, 20(123):1–26, 2019.
>
> [2] A. Wächter and L.T. Biegler. On the implementation of a primal-dual interior point filter line search algorithm for large-scale nonlinear programming. Mathematical Programming, 106(1):25–57, 2006
>
> [3] I. Bogunovic, A. Krause  Misspecified gaussian process bandit optimization. Advances in neural information processing systems. 2021;34:3004-15.
>
> [4] S Iwazaki, Improved Regret Bounds for Gaussian Process Upper Confidence Bound in Bayesian Optimization, arXiv:2506.01393, 2025•arxiv.org

---

> > ### Comment · Reviewer_3QQu · 2025-08-05
> > **Response to Rebuttal**
> >
> > I thank the authors for their careful consideration of my comments and for their detailed responses. I am quite satisfied with the responses and I'm glad the authors were able to strongly address my major weaknesses. In particular, I am delighted to see new enhancements to the theory that allow for constraint tolerance. In light of this as well as my reading of the other reviews, I am updating my evaluation/score to a 5.

---

> > > ### Author Response · Authors · 2025-08-05
> > > **Thank you reviewer 3QQu**
> > >
> > > We sincerely thank the reviewer for your valuable feedback and suggestions, which in our view helped strengthen the simple regret bound significantly. We are grateful for the increase in score and will incorporate all the changes discussed in the revised paper.

---

### Official Review · Reviewer_xe4v · 2025-07-02

**Clarity:** 3
**Significance:** 2
**Originality:** 3
**Rating:** 5
**Confidence:** 2

**Summary:**

This paper establishes the first theoretical convergence rates for the Constrained Expected Improvement (CEI) algorithm, a popular and widely used method for constrained Bayesian optimization (CBO), under both the frequentist and Bayesian settings. Despite its empirical success, the convergence rate of CEI had not been previously established.

The authors highlight that these results are not direct extensions from the unconstrained Expected Improvement (EI) analysis due to the complexities introduced by the constraints and the multiplicative nature of the CEI acquisition function. The paper also provides numerical experiments on both synthetic and benchmark problems to empirically validate the theoretical convergence results.

**Questions:**

1.  The analysis is performed in a noise-free setting. What new technical challenges would arise when extending this analysis to account for noisy observations of the objective and constraint functions, and how would you anticipate the convergence rates would be affected?
2. The CEI algorithm and the analysis assume that the objective and constraint functions are conditionally independent. How robust do you expect these convergence rates to be to violations of this assumption, which may occur in many practical engineering problems?

**Ethical Concerns:**

["NO or VERY MINOR ethics concerns only"]

**Final Justification:**

The major weaknesses are addressed by the authors' responses.

**Limitations:**

yes

**Quality:**

3

**Strengths And Weaknesses:**

Strengths:

1. This work is the first to establish the theoretical convergence rate for the Constrained Expected Improvement (CEI) algorithm, a widely used method for which such an analysis was previously missing.
2. The paper provides convergence rates under two different standard settings: a frequentist setting, where the functions are assumed to be in a Reproducing Kernel Hilbert Space (RKHS), and a Bayesian setting, where the functions are assumed to be sampled from Gaussian Processes (GPs).
3. The authors explicitly identify and address the technical challenges unique to analyzing CEI, such as the multiplicative structure of the acquisition function and the need to balance exploration for feasibility with exploitation for optimality.

Weaknesses:
1. The analysis is conducted for a noise-free setting, meaning the function evaluations are deterministic. This may not hold in many real-world black-box optimization problems where observations are noisy.
2. The CEI algorithm and the corresponding analysis assume that the objective and constraint functions are conditionally independent. For the extension to multiple constraints, it is also assumed that the constraints are conditionally independent of each other.

---

> ### Author Rebuttal · Authors · 2025-07-30
>
> We want to first thank the reviewer for taking your time and effort to review our paper. We appreciate your valuable feedback and hope we can sufficiently address them below.
>
> $\textbf{weakness 1}$ $\textit{Noise-free setting.}$
>
> R: We appreciate the reviewer’s comment regarding the assumption of noise-free observations and address it in two parts, corresponding to the objective and constraint functions, respectively.
>
> 1. The assumption of noise-free observations on the objective function $f$ stems from the lack of theoretical understanding of Expected Improvement (EI) in the noisy setting. To the best of our knowledge, even in the unconstrained case, no simple regret upper bound for noisy EI has been established. This difficulty arises primarily from the non-convexity of the EI acquisition function and the use of the noisy best observation in its formulation, which introduces significant analytical challenges. Given this unresolved issue in the unconstrained setting, extending the analysis to constrained EI (CEI) with noisy objective observations is particularly nontrivial.
>
> However, we have made an effort to partially address this challenge by allowing noise in the objective function observations while keeping constraint observations noise-free. Our analysis leverages a recent theoretical advancement that begins to bridge the gap for noisy EI in the unconstrained case. We have added the resulting simple regret bounds for CEI under this setting (noisy $f$ noise-free $c$) in the revised manuscript.
> In short, using a noisy simple regret definition, CEI achieves similar convergence rates in the Bayesian setting with i.i.d. Gaussian observation noise.
> The convergence rates are $\mathcal{O} (t^{-1/2}\log^{(d+1)/2}(t))$ for SE kernels and $  \mathcal{O}(t^{-\nu/(2\nu+d)} \log^{\nu/(2\nu+d)}(t))$ for Matérn kernels.
>
> 2. Regarding constraint observations, the requirement of noise-free evaluations is closely tied to the design of the CEI algorithm itself. In the presence of noise on the constraint function $c$, it becomes fundamentally ambiguous whether a given sample satisfies the constraint. This ambiguity complicates the definition of feasibility, as well as the identification of the best feasible solution—both of which are central to the CEI framework. Since our work focuses on a theoretical analysis of the original CEI algorithm, we follow the standard assumption of noise-free constraint observations.
>
> Nonetheless, to enhance the practical applicability of our results, we follow the suggestion of Reviewer 3QQu. We have included a constraint tolerance parameter $\epsilon>0$ so that practitioners can implement $c(x)\leq \epsilon$ to roughly account for small noise. This constraint formulation is commonly employed in the constrained Bayesian optimization literature [1] to accommodate small levels of noise. We have successfully extended our analysis to establish simple regret bounds under this relaxed feasibility definition and added it to the new results in the revised manuscript.
> While the upper bound of the simple regret changes with the inclusion of tolerance, the convergence rate remains the same.
>
> $\textbf{weakness 2}$ $\textit{Conditional independence of functions.}$
>
> R: The reviewer points out that the objective and constraints are required to be conditionally independent [2]. We would like to highlight that constrained Bayesian optimization methods often treat constraints independently by modeling each constraint with a separate surrogate (e.g., GP). When constraints are correlated, this assumption breaks down and tackling challenges in modeling joint constraint behavior is out of the scope of this paper. Since our focus is on providing a theoretical analysis of the original CEI algorithm [2], which inherently assumes independent objective and constraints, we have followed this assumption in our work. We plan to explore relaxing this assumption in future work.
>
> $\textbf{question 1}$ $\textit{Challenges for analysis in noisy setting.}$
>
> R: We appreciate this question. As we explained in our response to weakness 1, the extension to the noisy case is not accomplished even in the unconstrained case. The technical challenges are multiple. First, the current version of EI or CEI uses the best (feasible) observation as incumbent, which inherits the noise associated with that observation. This makes it very challenging to achieve a desirable convergence rate in the noisy case.
> Second, the EI function is nonlinear and non-convex, thus making its analysis more complicated.
>
> As we mentioned in weakness 1, we have newly derived convergence rates results under the noisy Bayesian setting and have updated the new results in the paper.
> However, the result is limited to noisy objective observations and the constraint observations are still required to be noise-free. The convergence rate on the noisy simple regret remains the same since we continue to use maximum information gain to establish the rates.
>
> If we hope to further analyze noisy constraint functions, we first need to define what counts as feasible in such a case. Since a point with $c(x)> 0$ can still be observed as feasible in the noisy case, we perhaps need a modification to the CEI algorithm to account for this noise.
> As an intermediate step, we have included a tolerance for the constraint, as we explained above in our response to weakness 1.
>
> $\textbf{question 2}$ $\textit{Convergence rate when constraints are correlated.}$
>
> R: Given its wide adoption in engineering publications, we would expect solid practical performance for CEI even when the conditional independence is violated.
> In terms of theoretical properties, we share some of our current understanding.
>
> We expect the simple regret bound to worsen given that the assumption does not accurately capture the relationship between constraints.
> Moreover, we speculate that a different simple regret bound is possible if we can quantify the correlation mathematically.
>
> We provide an intuitive example below.
> Consider two identical constraints $c_1(x)\leq 0$ and $c_2(x)=c_1(x)\leq 0$.
> Recall that the probability of feasibility (PoF) is $p_1(x)=\Phi(\frac{-\mu_t^{c_1}(x)}{\sigma_t^{c_1}(x)})<1$
> multiplying $p_2(x)=\Phi(\frac{-\mu_t^{c_2}(x)}{\sigma_t^{c_2}(x)})<1$. Thus, at a feasible point $x$, CEI would underestimate its probability of feasibility, which if modeled correctly would just be $p_1(x)$ instead of $p_1(x)p_2(x)$, especially initially. However, as the GP surrogate models of the constraints become more accurate, $p_1(x)$ and $p_2(x)$ both approach $1$ at a feasible point and its CEI value increases.
> Therefore, we anticipate the convergence of simple regret to hold but the upper bound to be worse.
>
> Finally, since the convergence rates are dominated by the maximum information gain of the objective, thanks to the multiplicative structure of CEI, it is possible that the convergernce rate itself could remain similar.
>
> [1] S. Ariafar, J. Coll-Font, D. Brooks, and J. Dy. ADMMBO: Bayesian optimization with unknown constraints using ADMM. Journal of Machine Learning Research, 20(123):1–26, 2019.
>
> [2] J. R. Gardner, M. J. Kusner, Z. Xu, K. Q. Weinberger, and J. P. Cunningham. Bayesian optimization with inequality constraints. In Proceedings of the 31st International Conference on International Conference on Machine Learning - Volume 32, ICML’14, pages II–937–II–945.

---

> > ### Comment · Reviewer_xe4v · 2025-08-06
> >
> > I thank the authors for the detailed responses. I will raise my score to 5.

---

> > > ### Author Response · Authors · 2025-08-07
> > > **Thank you reviewer xe4v**
> > >
> > > Thank you the reviewer for your response and raising the score. The questions you raised are all important to expected improvement and constrained expected improvement and we hope to continue to resolve some of them in future work. We are grateful for the valuable feedback.

---

### Official Review · Reviewer_DV68 · 2025-07-03

**Clarity:** 3
**Significance:** 3
**Originality:** 3
**Rating:** 5
**Confidence:** 3

**Summary:**

This paper derives the convergence rate of the constrained expected improvement (CEI) algorithm, a widely used method in constrained Bayesian optimization.
The analysis shows simple regret bounds for CEI on objective and constraint functions that lie in an RKHS and on those drawn from Gaussian processes.
The authors conducted experiments to confirm the results of the analysis.

**Questions:**

The analysis seems to assume that the number of constraint functions is one. However, I think there are several real-world problems that contain more than one constraint. Is it tedious work to extend the analysis result in such a situation?

I could not confirm how close the derived result was to the actual behavior of CIE from the result of the numerical experiment. Could you plot the derived convergence rate in Figures 1.a and 1.b (if possible)? In addition, please conduct experiments with different dimensions to confirm how it changes the convergence rate.

In Figures 2 and 3, I think the "infeasible" region is marked.

**Ethical Concerns:**

["NO or VERY MINOR ethics concerns only"]

**Final Justification:**

My concerns, including the limitation of the number of constraint functions and experimental validation of the derived results, have been resolved precisely. Therefore, I believe that the quality of the paper will be significantly improved in the revised version.

**Limitations:**

The analysis with more than 1 constraint function is desired in future work.

**Quality:**

3

**Strengths And Weaknesses:**

Strength:
This paper is the first work to derive the convergence rate of the CEI algorithm. In addition, the key points of the proof are summarized as proof sketches. These points increase the originality and clarity of this paper.

Weakness:
The numerical evaluation is not sufficient to check whether the derived convergence rate is reasonable.

---

> ### Author Rebuttal · Authors · 2025-07-30
>
> We want to first thank the reviewer for taking your time and effort to review our paper. We appreciate your valuable feedback and hope to sufficiently address them below.
>
> $\textbf{Weakness 1}$ $\textit{The numerical evaluation is not sufficient to check whether the derived convergence rate is reasonable.
> }$
>
> R: As the reviewer suggested, we have now performed additional numerical experiments to strengthen our results. Please see our response to question 2 for details.
>
> $\textbf{Question 1}$ $\textit{ Multiple constraints}$
>
> R: To clarify, we have already extended the results to problems with multiple constraints, as discussed in Remark 3.15. As noted there, this extension is possible under the assumption that the constraints are conditionally independent. This assumption is inherited from the original CEI method [1] and has been shown to yield strong empirical performance. In addition, we have empirically evaluated our algorithm on a problem with multiple constraints, and the results in Figure 1(d) demonstrate that it continues to perform well in such scenarios.
>
>
> $\textbf{Question 2}$ $\textit{Numerical experiment question and requests}$
>
>  The derived theoretical convergence rates depend on unknown problem-specific constants -- the maximum information gain, which prevents us from expressing the upper bounds in a fully explicit form. Also,  our theoretical analysis is asymptotic in nature.  As a result, it is not feasible to directly overlay the theoretical rates on our empirical plots for quantitative comparison. Nevertheless, the empirical results clearly exhibit sublinear convergence behavior, which qualitatively aligns with our theoretical predictions. This observed trend provides strong supporting evidence for the validity of our theoretical analysis, even though a direct quantitative comparison is not feasible.
>
> As the reviewer suggested, to further validate our theoretical findings,  we conducted additional experiments using objective and constraint functions sampled from both reproducing kernel Hilbert spaces (RKHSs) and Gaussian process priors defined by the squared exponential (SE) and Matérn kernels, across different input dimensions (specifically $d=2$ and $d=4$). The results are in Table 1 We plotted the log-log curve of simple regret against the number of iterations. In all cases, we consistently observed sublinear convergence patterns, which align well with our theoretical guarantees. These new empirical results have been included in the revised manuscript.
>
> Finally, we emphasize that our work provides the first theoretical convergence rate guarantee for CEI, establishing that its simple regret converges to zero. This is consistent with the results from our experiments.
>
> | Objective     | $t=0$              | $t=10$              |$t=20$              | $t=30$              |$t=40$              |$t=50$              |
> |-----------------|----------------|----------------|----------------|----------------|----------------|----------------|
> | RKSH, SE, $d=2$          | 0.170 | 0.008 | 0.003 | 0.003 | 0.003 |0.002     |
> | RKSH, SE, $d=4$          | 1.141 | 0.714 |0.116 | 0.034 | 0.026 | 0.024|
> | RKSH, Matern, $d=2$    | 0.192 | 0.006 | 0.005 | 0.004 | 0.001 | 0.001  |
> |RKSH, Matern, $d=4$     | 1.100 | 0.751 | 0.350 | 0.103 | 0.022 | 0.021 |
> |GP, SE, $d = 2$              | 0.756 | 0.154 | 0.027 | 0.008 | 0.008 | 0.004 |
> |GP, SE, $d=4$                | 1.584 | 0.966 | 0.614 | 0.358 | 0.352 | 0.323 |
> |GP, Matern, $d=2$         | 0.755 | 0.148 | 0.111 | 0.032 | 0.014 | 0.013 |
> |GP, Matern, $d=4$         |1.584 | 1.057 | 0.581 | 0.438 | 0.437 | 0.349 |
>
> Table 1: The simple regret of different functions vs the number of iterations.
>
>
> $\textbf{Question 3}$ $\textit{Infeasible region marked wrong}$
>
> R: Thank you for pointing this out. We have made the necessary correction in the revised version of this paper.
>
> [1] J. R. Gardner, M. J. Kusner, Z. Xu, K. Q. Weinberger, and J. P. Cunningham. Bayesian optimization with inequality constraints. In Proceedings of the 31st International Conferenceon International Conference on Machine Learning - Volume 32, ICML’14, pages II–937–II–945.

---

> > ### Comment · Reviewer_DV68 · 2025-08-04
> >
> > Thank you for the detailed response. I will raise my score to 5. Please incorporate the contents in the response into their manuscript or supplementary material.

---

> > > ### Author Response · Authors · 2025-08-04
> > > **Thank you reviewer for the update**
> > >
> > > We sincerely thank the reviewer for the updated evaluation and positive feedback. We will incorporate the revised content into the manuscript and supplementary material.

---

### Official Review · Reviewer_QM2L · 2025-07-20

**Clarity:** 3
**Significance:** 3
**Originality:** 4
**Rating:** 5
**Confidence:** 4

**Summary:**

This paper provides a theoretical convergence analysis of the Constrained Expected Improvement (CEI) algorithm in Constrained Bayesian Optimization. It establishes simple regret bounds under both frequentist and Bayesian settings, showing improved convergence rates for squared exponential and Matérn kernels. Using information-theoretic tools, the analysis overcomes challenges from CEI’s multiplicative structure. Results are supported by experiments, offering theoretical justification for CEI’s practical success.

**Questions:**

1. How does the analysis handle the non-convexity and potential multimodality of the CEI acquisition function in practice? The regret bounds assume exact maximization of CEI at each iteration, but the acquisition function is non-convex and often only approximately optimized in practice. How sensitive are the theoretical guarantees to optimization error in the acquisition step?

2.  How valid is the assumption of conditional independence between the objective and constraint GPs in real applications? The CEI analysis assumes conditional independence, yet in practice, objective and constraint functions may be correlated. How would the theoretical results change if this assumption were relaxed or violated?

3.  Can the authors clarify how the feasibility probability is controlled in the regret analysis? The bound includes inverse terms like $1 / \Phi(-B_c)$, which can be large or unstable if the feasibility boundary is close to zero. How robust is the bound when $c(x^*) \approx 0$, i.e., the optimal point lies near the constraint boundary?

4. Is there a path to extending the current analysis to noisy or approximate observation settings? Given that real-world evaluations are often noisy, can the authors discuss how their framework could be generalized to include noise, possibly following techniques used in noisy GP-UCB or Thompson Sampling literature?

5. How does the multiplicative acquisition function structure affect the exploration-exploitation trade-off compared to additive or penalty-based methods? The CEI acquisition function is the product of EI and feasibility probability. How does this structure influence the theoretical exploration behavior, especially in high-dimensional or narrow-feasibility domains?

**Ethical Concerns:**

["NO or VERY MINOR ethics concerns only"]

**Final Justification:**

The authors provided a detailed response to my comments, which is fairly convincing. I am glad to raise my score to 5.

**Quality:**

3

**Strengths And Weaknesses:**

strengths:
1. This is the first work to derive simple regret upper bounds for the CEI algorithm under both frequentist and Bayesian settings.
2. Under frequentist assumptions (RKHS model), they derive improved rates for SE kernel $\mathcal{O}(t^{-1/2} \log^{(d+1)/2}(t))$ and for Matérn kernel  $\mathcal{O}(t^{-\nu/(2\nu+d)} \log^{\nu/(2\nu+d)}(t))$. These rates improve over the direct constrained extension of Bull (2011), which achieved $\mathcal{O}(t^{-1/d})$.
3. The authors show that, with high probability, the same convergence rates hold when f and c are samples from GPs. This complements prior work like Srinivas et al. (2009) for unconstrained GP-UCB, extending it to CEI.
4. The paper leverages maximum information gain bounds to obtain tighter regret bounds, following approaches in recent literature on UCB (e.g., Lyu et al., 2019; Vakili et al., 2021).

weaknesses:
1. The analysis requires the existence of an initial feasible sample, which is not always easy to guarantee in practice. Without it, the CEI acquisition function is undefined because the improvement is computed over the best feasible value.
2. The convergence analysis is conducted only under a noise-free setting, where evaluations of both the objective and constraint functions are assumed to be exact.
3. The analysis assumes a single constraint (or extends to multiple constraints under independence assumptions). However, real-world CBO often involves correlated or conflicting constraints, which are not addressed.
4. There’s no non-asymptotic regret bound (e.g., PAC-style guarantees) to quantify performance in small-budget settings.

---

> ### Author Rebuttal · Authors · 2025-07-30
>
> We sincerely thank the reviewer for taking the time and effort to review our paper. We greatly appreciate your valuable feedback and have made every effort to address your comments thoroughly in the detailed responses below.
>
> $\textbf{weakness 1}$ $\textit{The analysis requires the existence of an initial feasible sample, which is not always easy to guarantee in practice. }$
>
> R: We appreciate the reviewer’s insightful observation. As noted in Remark 3.1 of our paper, this indeed represents a fundamental limitation of the Constrained Expected Improvement (CEI) algorithm originally proposed in [1]. Our primary objective in this work is to provide theory to this widely used algorithm, rather than to introduce modifications or improvements. Accordingly, this limitation is not only acknowledged but also  incorporated into our theoretical framework.
>
> Some recent works have tried to address the issue of infeasible initial samples. One common strategy focuses on finding an initial feasible sample before switching back to CEI. For instance, in [2] the authors proposed using only the GP of the constraint function first, until a feasible initial sample is identified. In this case, our analysis still holds as the number of iterations increases and feasible samples become available.
>
> $\textbf{weakness 2 and question 4}$ $\textit{The result is in noise-free setting}$
>
> R: We hope to address this comment in two parts, corresponding to the objective and constraint functions, respectively.
>
> 1) The assumption of noise-free observations on the objective function $f$ stems from the lack of theoretical understanding of Expected Improvement (EI) in the noisy setting. To the best of our knowledge, even in the unconstrained case, no simple regret bound for noisy EI has been established. This difficulty arises primarily from the non-convexity of the EI acquisition function and the use of the noisy best observation in its formulation, which introduces significant analytical challenges. Given this unresolved issue in the unconstrained setting, extending the analysis to constrained EI (CEI) with noisy objective observations is particularly nontrivial.
>
>   However, we have made an effort to partially address this challenge by allowing noise in the objective function observations while keeping constraint observations noise-free. Our analysis leverages a recent theoretical advancement that begins to bridge the gap for noisy EI in the unconstrained case. We would be pleased to include the resulting simple regret bounds for CEI under this setting (noisy $f$ noise-free $c$) in the revised manuscript. The convergence rates are $\mathcal{O} (t^{-1/2}\log^{(d+1)/2}(t))$ for SE kernels and $  \mathcal{O}(t^{-\nu/(2\nu+d)} \log^{\nu/(2\nu+d)}(t))$ for Matérn kernels.
>
> 2. Regarding constraint observations, the requirement of noise-free evaluations is closely tied to the design of CEI itself. In the presence of noise on the constraint function $c$, it becomes fundamentally ambiguous whether a given sample satisfies the constraint. This ambiguity complicates the definition of feasibility, as well as the identification of the best feasible solution—both of which are central to the CEI framework. Since our work focuses on a theoretical analysis of CEI, we follow the standard assumption of noise-free constraint observations.
>
> Nonetheless, to enhance the practical applicability of our results, we follow the suggestion of Reviewer 3QQu. We include a constraint tolerance  $\epsilon>0$ so that practitioners can implement $c(x)\leq \epsilon$ to roughly account for small noise. This constraint formulation is commonly employed in the constrained Bayesian optimization literature [4] to accommodate small levels of noise. We have successfully extended our analysis to establish simple regret bounds under this relaxed feasibility definition and have included the new results in the revised manuscript. The convergence rates remain the same.
>
> 3. In addition, GP-UCB and TS techniques do not apply to EI because EI has the noisy observation directly used in its formula (the best observation) and is highly nonlinear and nonconvex.
>
> $\textbf{weakness 3 and question 2}$ $\textit{Conditional independence of constraints}$
>
> R: We agree that correlated constraints are more realistic in real-world applications. However, we would like to point out that
> this assumption is made by the CEI algorithm itself. Given its wide adoption and strong empirical performance, we anticipate CEI to work relatively well with a limited number of constraints that are not conditionally independent. In line with these established precedents, we have followed this assumption in our work. We plan to explore relaxing this assumption in future work.
>
> We briefly provide some intuition below about how this assumption would impact the convergence of CEI. Consider two identical constraints $c_1(x)\leq 0$ and $c_2(x)=c_1(x)\leq 0$. Recall that the probability of feasibility (PoF) is $p_1(x)=\Phi(\frac{-\mu_t^{c_1}(x)}{\sigma_t^{c_1}(x)})<1$ multiplying $p_2(x)=\Phi(\frac{-\mu_t^{c_2}(x)}{\sigma_t^{c_2}(x)})<1$. Thus, at a feasible point $x$, CEI would underestimate its probability of feasibility, which if modeled correctly would just be $p_1(x)$ instead of $p_1(x)p_2(x)$, especially initially. However, as the GP surrogate models of the constraints become more accurate, $p_1(x)$ and $p_2(x)$ both approach $1$ at a feasible point and its CEI value increases. Therefore, we anticipate the regret bound to worsen but convergence continues to hold. Further, since the convergence rates are dominated by the maximum information gain of the objective, thanks to the multiplicative structure of CEI, it is possible that the convergence rate itself remains similar when the condition is violated.
>
> $\textbf{weakness 4}$ $\textit{No non-asymptotic bound}$
>
> Thank you for this insightful comment. We are able to fully address this concern. To the best of our knowledge, there are very few existing results on non-asymptotic regret bounds in the BO literature. From our Theorem 3.8 and 3.12, we can derive a new result that establishes the iteration complexity—i.e., the number of iterations required to achieve a given simple regret error $\epsilon$. For instance, to reach an error of $\epsilon$ we need $\epsilon^{-2} \log^{-(d+1)}(1/\epsilon)$ iterations. We have added this theorem along with its proof into the revised manuscript.
>
> $\textbf{Question 1}$ $\textit{Exact solution to EI}$
>
> Thank you for this insightful question. We first answer how to find maximum of CEI in practice. A common technique is to use multiple starting points and gradient-based methods to find better solutions. For unconstrained EI, there has also been work that suggests a branch and bound approach can solve the maximization of EI relatively well [4]. As the reviewer stated, in practice often only an approximate maximum CEI can be found. We totally understand this point and it is in fact a general issue for the majority of BO algorithms. Therefore, resolving this issue is beyond the scope of our paper.
>
> As for the sensitivity of regret bounds to the optimization error in the acquisition step, this itself is an active area of research (a recent paper on this is [5]). Although addressing this problem is beyond the scope of this paper, we would be happy to share our perspectives on this topic if the reviewer is interested.
>
> $\textbf{Question 3}$ $\textit{Robustness at boundary}$
>
> The constant $B_c$ is the RKHS norm of function c and does not change because of $c(x^*) = 0$.
>
> The probability of feasibility is controlled via the confidence interval of the Gaussian process, i.e., $|\mu_t(x)-c(x)|\leq B_c \sigma_t(x)$. If $c(x)=0$, we have  $|\mu_t(x)|\leq B_c \sigma_t(x)$. Thus, as $\sigma_t(x)$ decreases, $|\mu_t(x)|$ decreases as well, making the probability of feasibility $\Phi(\frac{-\mu_t(x)}{\sigma_t(x)})$ controllable. The confidence interval and the regret bound holds for $c(x^*)=0$.
>
> $\textbf{Question 5}$ $\textit{Multiplicative structure of CEI}$
>
> In our experience, the multiplicative structure forces more exploration around the feasible region predicted by the GP models due to the quick decaying of the PoF function $\Phi(\cdot)$.
> It also mitigates the burden associated with hyperparameter tuning compared to penalty-based methods since one does not need to tune the penalty parameter.
> In theory, an additive structure can induce an additive regret bound with contribution from both the objective and constraint and likely the penalty parameter. In contrast, the constraint affects our regret bound by an additional multiplicative factor.
>
> We add our regret bounds hold for high-dimensional problems or narrow-feasibility domains and we do observe good CEI performance in our own experience for medium-dimensional problems and problems with small feasible regions.
>
> [1] J. R. Gardner, M. J. Kusner, Z. Xu, K. Q. Weinberger, and J. P. Cunningham. Bayesian optimization with inequality constraints. In Proceedings of the 31st International Conference on International Conference on Machine Learning - Volume 32, ICML’14,b JMLR.org, 2014.
>
> [2] Q. Lin, J. Hu, Q. Zhou, L. Shu, A. Zhang. A multi-fidelity Bayesian optimization approach for constrained multi-objective optimization problems, Journal of Mechanical Design, 2024
>
> [3] S. Ariafar, J. Coll-Font, D. Brooks, and J. Dy. ADMMBO: Bayesian optimization with unknown constraints using ADMM. Journal of Machine Learning Research, 20(123):1–26, 2019.
>
> [4] A. M. Schweidtmann, D. Bongartz,  D. Grothe, T. Kerkenhoff, X. Lin, J. Najman, and A. Mitsos, 2021. Deterministic global optimization with Gaussian processes embedded. Mathematical Programming Computation, 13(3), pp.553-581.
>
> [5] H. Kim, C. Liu, and Y. Chen. Bayesian Optimization with Inexact Acquisition: Is Random Grid Search Sufficient? rXiv:2506.11831, 2025.

---

### Decision · Program_Chairs · 2025-09-17

**Decision:**

Accept (spotlight)

**Comment:**

[DRAFT NOTES]

Shows convergence rates for constrained expected improvement when the objective function is in an RKHS and with high probability when the objective function is drawn from a Gaussian process.

Strengths
- Significance and novelty --- this the first theoretical convergence rate for an important algorithm
- Difficulty of proving the result
- Clarity

Weakness
- Only handles noise-free
- Assumes first point is feasible

I recommend acceptance and a highlight poster because of the significance of this work. This is an old problem that many people have tried to solve.